# Reconstruct high-resolution 3D genome structures for diverse cell-types using FLAMINGO

Hao Wang[1], Jiaxin Yang[1], Yu Zhang[2], Jianliang Qian [1,3✉] & Jianrong Wang [1✉]

High-resolution reconstruction of spatial chromosome organizations from chromatin contact maps is highly demanded, but is hindered by extensive pairwise constraints, substantial missing data, and limited resolution and cell-type availabilities. Here, we present FLAMINGO, a computational method that addresses these challenges by compressing inter-dependent Hi-C interactions to delineate the underlying low-rank structures in 3D space, based on the low-rank matrix completion technique. FLAMINGO successfully generates 5 kb- and 1 kb-resolution spatial conformations for all chromosomes in the human genome across multiple cell-types, the largest resources to date. Compared to other methods using various experimental metrics, FLAMINGO consistently demonstrates superior accuracy in recapitulating observed structures with raises in scalability by orders of magnitude. The reconstructed 3D structures efficiently facilitate discoveries of higher-order multi-way interactions, imply biological interpretations of long-range QTLs, reveal geometrical properties of chromatin, and provide high-resolution references to understand structural variabilities. Importantly, FLAMINGO achieves robust predictions against high rates of missing data and significantly boosts 3D structure resolutions. Moreover, FLAMINGO shows vigorous cross cell-type structure predictions that capture cell-type specific spatial configurations via integration of 1D epigenomic signals. FLAMINGO can be widely applied to large-scale chromatin contact maps and expand high-resolution spatial genome conformations for diverse cell-types.

---

[1] Department of Computational Mathematics, Science and Engineering, Michigan State University, East Lansing, MI 48824, USA. [2] Center for Immunobiology, Department of Investigative Medicine, Western Michigan University Homer Stryker M.D. School of Medicine, Kalamazoo, MI 49007, USA. [3] Department of Mathematics, Michigan State University, East Lansing, MI 48824, USA. ✉email: jqian@msu.edu; wangj164@msu.edu

The three-dimensional (3D) architecture of genomes plays pivotal roles in DNA replication, genome stability, and tissue differentiation[1–3]. Quantitative characterization of spatial chromosome conformations is crucial for deciphering the complex systems of spatially coordinated transcriptional and epigenetic activities[4–6], leading to the understanding of gene regulation mechanisms. The genome-wide high-throughput chromosome conformation capture technique such as Hi-C[7,8] has been one of the driving forces in studies of 3D genome structures. The Hi-C datasets profiled from different cell-types and species[7,9–13] have revealed structural components of genome organization[7,10,14], such as chromatin loops, topologically associated domains (TADs), and chromatin compartments. Although these findings have provided powerful insights into the governing rules of chromosome folding at large scales (~100 kb–1 Mb), such as the loop extrusion model[15,16], it is still computationally difficult to accurately reconstruct high-resolution spatial conformations, such as at ~5 kb resolution, for all chromosomes in large genomes.

Since the collection of Hi-C experiments is growing, the resulting massive Hi-C data call for efficient computational algorithms for modeling 3D genomes. Previous algorithms of 3D reconstruction using Hi-C data have been able to predict spatial distances mainly at low-resolutions or within specific genomic segments[17]. Typically, based on experimentally estimated conversion functions[14], the observed Hi-C contact frequency is converted into spatial distances, which we term as observed Hi-C distances in this paper. In general, a consensus structure or an ensemble of structures are inferred by maximizing the similarity between predicted and observed Hi-C distances using optimization-based (such as MDS-type or manifold learning techniques)[18–27] or probabilistic approaches (such as MCMC strategy)[28–32]. Representative state-of-the-art algorithms that have been shown to outperform other methods, along with some recent developments, include ShRec3D[33], GEM-FISH[34], Hierarchical3DGenome[35], RPR[36], SuperRec[37], ShNeigh[38], and PASTIS[28] (Methods section, Supplementary Note 1). The accuracy of a predicted structure is mainly evaluated by its capability of recapitulating the measured pairwise distances between genomic loci from Hi-C. Spearman correlation is one of the widely used metrics to quantify the accuracy. However, four fundamental challenges still need addressing in developing an efficient algorithm: (1) high scalability to reconstruct high-resolution spatial configurations for all chromosomes from massive Hi-C datasets; (2) superior performance to handle large fractions of missing data, which is a common drawback of Hi-C experiments; (3) capability to make accurate cross cell-type structure predictions, since the vast majority of cell-types lack Hi-C data; and (4) capability to predict high-resolution structures from low-resolution Hi-C contact maps.

To address the above four challenges, we have developed a low-rank matrix completion based methodology for reconstructing 3D genome structures from Hi-C data. Low-rank matrix completion has been found to be a powerful modeling framework for 3D shape inferences in different scientific fields[39–41]. One of the key advantages of such a modeling method is that it is able to explicitly leverage the low-rank property of a pairwise-distance matrix (rank ≤5 for Euclidean distance matrix, see Methods section)[42] in an objective function for optimization, and such a low-rank property has not been explicitly utilized in previous approaches, such as multidimensional scaling based methods. Efficient incorporation of the low-rank constraint into the modeling process allows fast structure reconstruction from just a small subset of Hi-C data, making the algorithm scalable for high-resolution structure predictions for large chromosomes with high fractions of missing data.

Our efforts have led us to create a Fast Low-rAnk Matrix completion algorithm for reconstructINg high-resolution 3D Genome Organizations from Hi-C data, FLAMINGO (https://github.com/wangjr03/FLAMINGO), which has been implemented to generate both 5 kb- and 1 kb-resolution 3D chromosomal structures for the human genome. Based on extensive performance evaluations using data from both simulated structures and experimental Hi-C datasets from the human genome, the high-resolution chromosome structures generated by FLAMINGO demonstrate substantially improved accuracy, compared with other state-of-the-art methods. The predicted high-resolution spatial distances in 3D space are further justified by orthogonal experiments (such as ChIA-PET[43], Capture-C[44,45], and SPRITE[46]), providing biological insights into long-range chromatin interactions in gene regulation. Beyond 2D contact maps, the predicted 3D structures by FLAMINGO can help to identify higher-order multi-way chromatin interactions, interpret potential mechanisms of genetic QTLs, characterize the geometrical patterns of chromatin folding, and facilitate the understandings of structural variations. Moreover, even using only 10% of down-sampled Hi-C contacts, FLAMINGO still achieves higher accuracy than other methods, demonstrating its superior capability of handling missing data in Hi-C. In addition, an integrative version of our algorithm, iFLAMINGO, is built to further combine 1D epigenomics data, such as DNase-seq signals, with Hi-C data, which allows us to make cross cell-type predictions of 3D genome architectures and boost the resolution of predictions. These algorithmic advantages will not only expand the coverage of cell-types for 3D genome modeling but also improve the information extraction from the fast-growing collection of experimental Hi-C data.

## Results

**FLAMINGO algorithm to reconstruct high-resolution 3D genome architectures.** Based on the 'beads on a string' polymer model[47], every chromosome is modeled as a chain of 'beads' consisting of DNA fragments or loci, and the pairwise distances between genomic loci are biologically induced from the Gram matrix of their 3D coordinates (Fig. 1a). To reconstruct the 3D spatial structure, the normalized chromatin contact maps from Hi-C experiments can be converted into an observed distance matrix as suggested by previous studies[10,14], whose validity and robustness are justified by both computational model selections and empirical comparisons with image-based data (see Methods section). The observed distance matrix typically contains large portions of unmeasured distances (namely, missing data), especially for high-resolution genomic loci (~5 kb fragments)[10]. FLAMINGO predicts the optimal genome structure based on a low-rank matrix completion framework (Fig. 1a and Methods section). The objective function contains three terms: (1) a term to impose the low-rank constraint on the Gram matrix of predicted 3D coordinates, since the 3D distance matrix has a rank at most five; (2) a term measuring the differences between predicted and observed distances, which is evaluated on the measured subset of pairs of loci; and (3) a penalty term penalizing unrealistic distances between adjacent DNA fragments. FLAMINGO uses the alternating-direction method of multipliers[48] to solve the optimization problem. At convergence, the optimal 3D structure that minimizes the objective function is identified, along with the completed pairwise distance matrix (Fig. 1a).

The key feature of FLAMINGO is to incorporate the low-rank constraint (rank ≤ 5) of the 3D distance matrix of size $N \times N$ into the optimization process, where $N$ is the number of genomic loci, such as the number of 5 kb DNA fragments. Since the pairwise spatial distances are generated by the 3D coordinate matrix of genomic loci (rank ≤ 3), the resulting symmetric Euclidean distance matrix has a rank at most 5[42]. It is because the squared

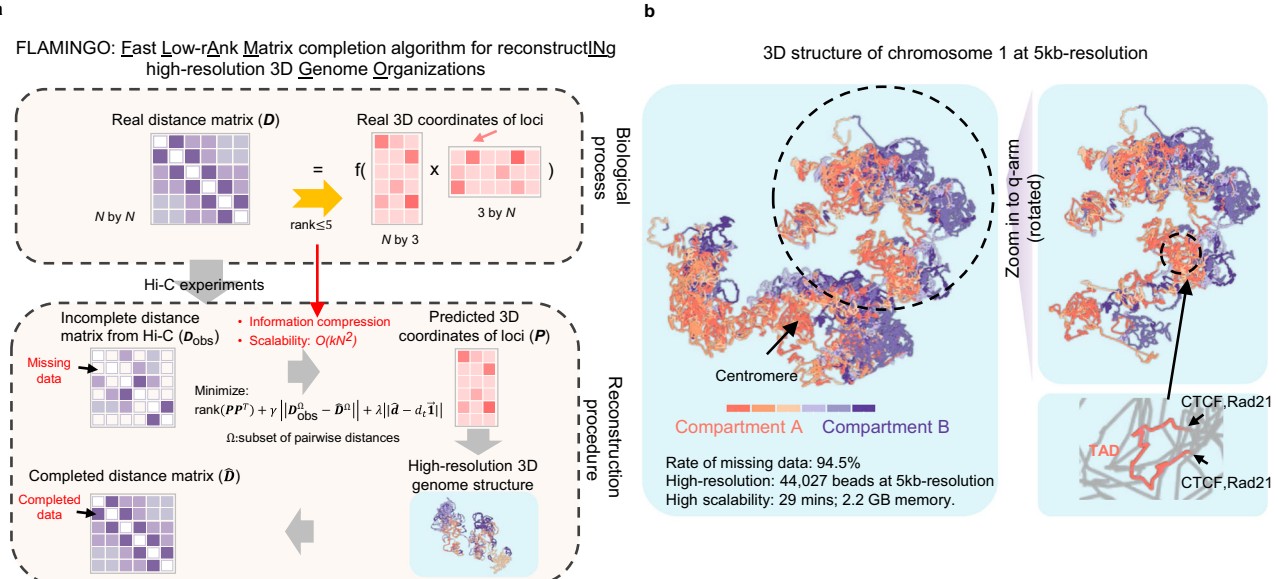

**Fig. 1 FLAMINGO reconstructs high-resolution 3D genome structures from Hi-C data. a** Schematic figure of FLAMINGO. Biologically, the distance matrix (size $N$ by $N$) is induced by the 3D coordinate matrix of DNA fragments (size $N$ by 3), which guarantees that the rank of the distance matrix is no more than five (upper panel). The low-rank property suggests the potential of information compression ($N^2$ entries to $5N$ entries), and enables FLAMINGO to efficiently reconstruct structures from incomplete distance matrices and perform superiorly against large portions of missing data. Equipped with high scalability, FLAMINGO can quickly predict the optimal coordinate matrix that reproduces the observed distances from Hi-C data (middle panel), leading to the high-resolution 3D genome structure and the completed distance matrix (lower panel). **b** Reconstructed 5 kb-resolution structure of chromosome 1 in the human genome by FLAMINGO. Chromatin compartments (A: orange; B: blue) demonstrate polarized positioning in the predicted structure. A representative example of predicted loop structures is shown in the zoom-in view, where both anchors interact with each other (supported by ChIA-PET interactions) and are bound by CTCF and Rad21. Color gradients represent consecutive TADs within each type of compartments.

Euclidean distance matrix is a sum of three matrices: one being the Gram matrix of rank at most 3 and each of the other two being of rank at most 1 (see Methods section). And thus, it has intrinsic degrees of freedom at most $5N$, which, compared to the size $N \times N$ of the entire full matrix, is extremely small when $N$ is large. Therefore, in order to recover the entire distance matrix, we may just need the number of measurements of the distance matrix to be proportional to the intrinsic degrees of freedom. In fact, as long as the information of the underlying distance matrix is not concentrated on a few entries, each randomly selected measurement of pairwise distances will be equally informative, suggesting that the information can be substantially compressed[49] (Fig. 1a). Hence, by minimizing the rank of the inferred Gram matrix, low-rank matrix completion models[49] offer at least two benefits (Methods section): (1) accurate 3D structures can be reconstructed from subsets of observed distances; and (2) fast matrix calculations can be carried out based on sparsity and low-rankness of the underlying matrices. Remarkably, both benefits are heavily needed for high-resolution structure predictions. By dividing the genome into high-resolution DNA fragments such as at 5 kb-resolution, the size of the distance matrix becomes huge, many entries of which have no data due to the limited sequencing depth of Hi-C experiments. Thus, FLAMINGO is able to build high-resolution 3D structures from the fast-growing collection of Hi-C datasets with decent scalability at computational complexity $O(N^2)$ without demanding increased sequencing depths (Methods section, Supplementary Note 1, and Supplementary Figs. 1 and 2).

To enable parallel computations, FLAMINGO also employs a hierarchical strategy by dividing each chromosome into 1 Mb domain-level fragments that are further divided into 5 kb DNA fragments, where we define a 1 Mb fragment as a domain (Methods). The same low-rank matrix completion algorithm is applied on both the inter-domain hierarchy consisting of 1 Mb fragments, which leads to a basic structural skeleton, and the

intra-domain hierarchy of 5 kb fragments, which results in intra-domain structures. Different from other methods that only align the endpoints of domain fragments[34] or whose refinement processes are dominated by intra-domain distances[35], an iterative rotation algorithm along the three spatial directions is developed to assemble intra-domain structures into the inter-domain skeleton, by aligning all measured off-diagonal distances so as to maximize the consistency with inter-domain 5 kb-resolution Hi-C contacts (Supplementary Fig. 3 and Methods section). At convergence, the iterative rotation algorithm leads to the full high-resolution structures for each chromosome.

FLAMINGO has been applied on the normalized Hi-C datasets from six human cell-types (GSE63525[10]) to generate 3D structures for chromosomes 1–22 and X at 5 kb-resolution (Supplementary Fig. 1), which are the largest resources of reconstructed 3D structures for the human genome at high-resolution (https://github.com/wangjr03/FLAMINGO). For example, at 5 kb-resolution, chromosome 1 contains 44,027 DNA fragments, excluding the centromere and telomere regions, and 94.5% entries of the observed distance matrix in GM12878 are missing data. The structure of chromosome 1 can be predicted quickly by FLAMINGO (Fig. 1b). The two types of chromatin compartments (A/B) are organized into separable positions in the predicted structure, consistent with the polarized architecture observed from the multiplexed FISH[14]. By zooming into the high-resolution structure, predicted loop structures are found corresponding to previously annotated TADs (Fig. 1b), where the pairs of CTCF-associated Hi-C loop anchors (CTCF-CTCF pairs) are predicted with significantly shorter spatial distances, compared to genomic-distance controlled pairs in two cases: (1) pairs between a CTCF-anchor and a random anchor with the same genomic separation (CTCF-random pairs, Supplementary Fig. 1 boxplot, right, $p$-value $= 5.21 \times 10^{-4}$, one-sided Wilcoxon test), and (2) pairs between random anchors with the same genomic separation (random-random pairs, Supplementary Fig. 1 boxplot,

left, $p$-value $= 2.78 \times 10^{-5}$, one-sided Wilcoxon test). In addition, FLAMINGO has also generated 3D chromosomal structures at 1 kb-resolution in GM12878 for all chromosomes (Supplementary Fig. 2), which represent spatial reconstructions with the highest resolution to date. Moreover, FLAMINGO is robust to the choice of conversion factors for converting interaction frequency to distance, where the conversion factor is chosen within the range suggested by previous studies[14,28] (Supplementary Fig. 4).

**Benchmark performance based on simulated structures**. The performance of FLAMINGO was benchmarked on simulated structures. The distance matrix generated from the benchmark structure was randomly down-sampled and then mixed with noise (Fig. 2a and Methods section). By applying FLAMINGO on the noisy incomplete distance matrices, the reconstructed 3D structures can be identified with fast convergence (Supplementary Fig. 5), and they are in strong agreement with the original benchmark structures (relative error < 0.03 and correlation > 0.999) (Fig. 2b). In addition, the accuracy is robust against a

wide range of down-sampling rates and different levels of noise (Fig. 2c, d and Supplementary Fig. 5, correlation > 0.999), demonstrating that FLAMINGO is capable of handling missing data. The high accuracy is also found to be robust when FLAMINGO is applied to a series of simulated structures with different sizes (Supplementary Fig. 6), suggesting the performance is not affected by the number of genomic loci along chromosomes. Furthermore, to validate the iterative assembly algorithm for organizing intra-domain structures, we partitioned the benchmark structure into different domains and then reconstructed the whole structure using the assembly algorithm. The assembled structures recapitulate the benchmark structure with high accuracy (relative error < 0.005, correlation > 0.999) and are independent of specific choices of domain partitions (Supplementary Fig. 7a, b).

**Superior reconstruction accuracy across diverse cell-types**. The performance of FLAMINGO on experimental Hi-C data in the human genome was then systematically evaluated and compared with the state-of-the-art methods. As demonstrated in Fig. 1b and Supplementary Fig. 1, FLAMINGO is able to quickly reconstruct 3D chromosome structures at 5 kb-resolution, which are qualitatively consistent with both large-scale chromatin properties, such as compartments and TADs, and small-scale structural details, such as chromatin loops and CTCF/cohesin bindings. The predicted structural skeletons (1Mb-resolution) of chromosomes are strongly supported by results from both Hi-C (average correlation = 0.95, Supplementary Fig. 8a) and FISH[14] (average correlation = 0.80, Supplementary Fig. 8b and Supplementary Note 1), consistently higher than other methods. The reconstructed structures also vary across different cell-types, consistent with cell-type-specific chromatin contact patterns from Hi-C (Supplementary Fig. 9). Taking the predicted structure of chromosome 21 in GM12878 as an example, FLAMINGO reconstructs clear loop structures for TADs and predicts short 3D distances for inter-TAD chromatin contacts (Fig. 3a). Compared to the fuzzy input distance matrix converted from Hi-C (Supplementary Fig. 10), the distance matrix derived from the predicted 3D structure shows substantially improved resolution (Fig. 3a), and the reconstructed long-range inter-TAD contacts are supported by experimental Capture-C interactions (Supplementary Fig. 10).

To quantitatively evaluate the genome-wide accuracy at 5 kb-resolution, the predicted 3D chromosome structures were evaluated according to their consistency with the observed distance matrices derived from Hi-C (Methods section). Similarities between structures are quantified by Spearman correlations, which have been widely used as accuracy metrics in structure analysis. To note, achieving high correlations at 5 kb-resolution is a much harder problem than at low-resolutions (e.g. 100 kb- or 1 Mb-resolution), because Hi-C signals at 5 kb-bins are much noisier and the number of high-resolution constraints in optimization is huge. Remarkably, the Spearman correlations between the predicted and observed 3D distances at 5 kb-resolution, including both diagonal sub-matrices for intra-domain structures and off-diagonal sub-matrices for inter-domain structures, are robustly high across all six cell-types (Fig. 3b, left). The predicted structure in IMR90 shows the highest correlation (average correlation = 0.603 across 23 chromosomes), followed by structures predicted in GM12878 and K562 (average correlations = 0.512 and 0.525 respectively). The Spearman correlations based on off-diagonal points alone (i.e. inter-domain distances) also show similar levels (correlations > 0.42), except for HUVEC (correlation = 0.32). These results are significant achievements, considering the extensive noisy constraints imposed by the huge number of pairwise distances at 5 kb-resolution. For example,

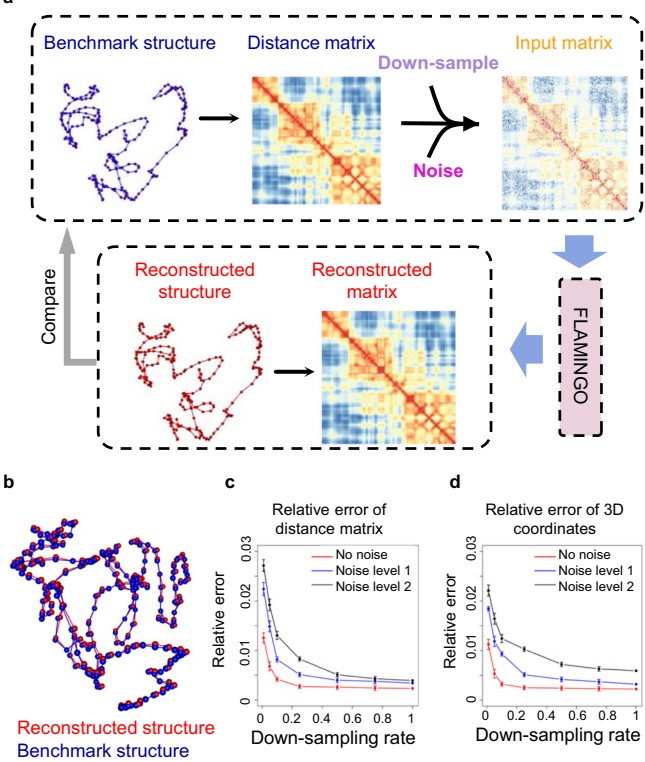

**Fig. 2 Performance evaluation based on simulations. a** Given a benchmark structure, the distance matrix is down-sampled using different down-sampling rates and mixed with different levels of noise (Noise level 1: low-level; Noise level 2: high-level; see Methods). The incomplete noisy distance matrices are used as inputs for FLAMINGO. The reconstructed 3D structures are compared with the benchmark structure by calculating relative errors and correlations. **b** One example of the reconstructed structure by FLAMINGO (down-sampling rate = 0.5, noise level 1, see Methods), which aligns with the benchmark structure almost identically (correlation = 0.9999999, relative error = 0.0037). **c, d** The performance of FLAMINGO (relative errors: the $y$-axis) under various down-sampling rates and noise levels, with respect to the accuracy of 3D distance matrices (**c**) and 3D coordinates of DNA fragments (**d**). Error bars represent the standard deviations of relative errors examined based on $n = 10$ independently down-sampled distance matrices under each down-sampling rate. Data are presented as mean values ±SD. Source data are provided as a Source Data file.

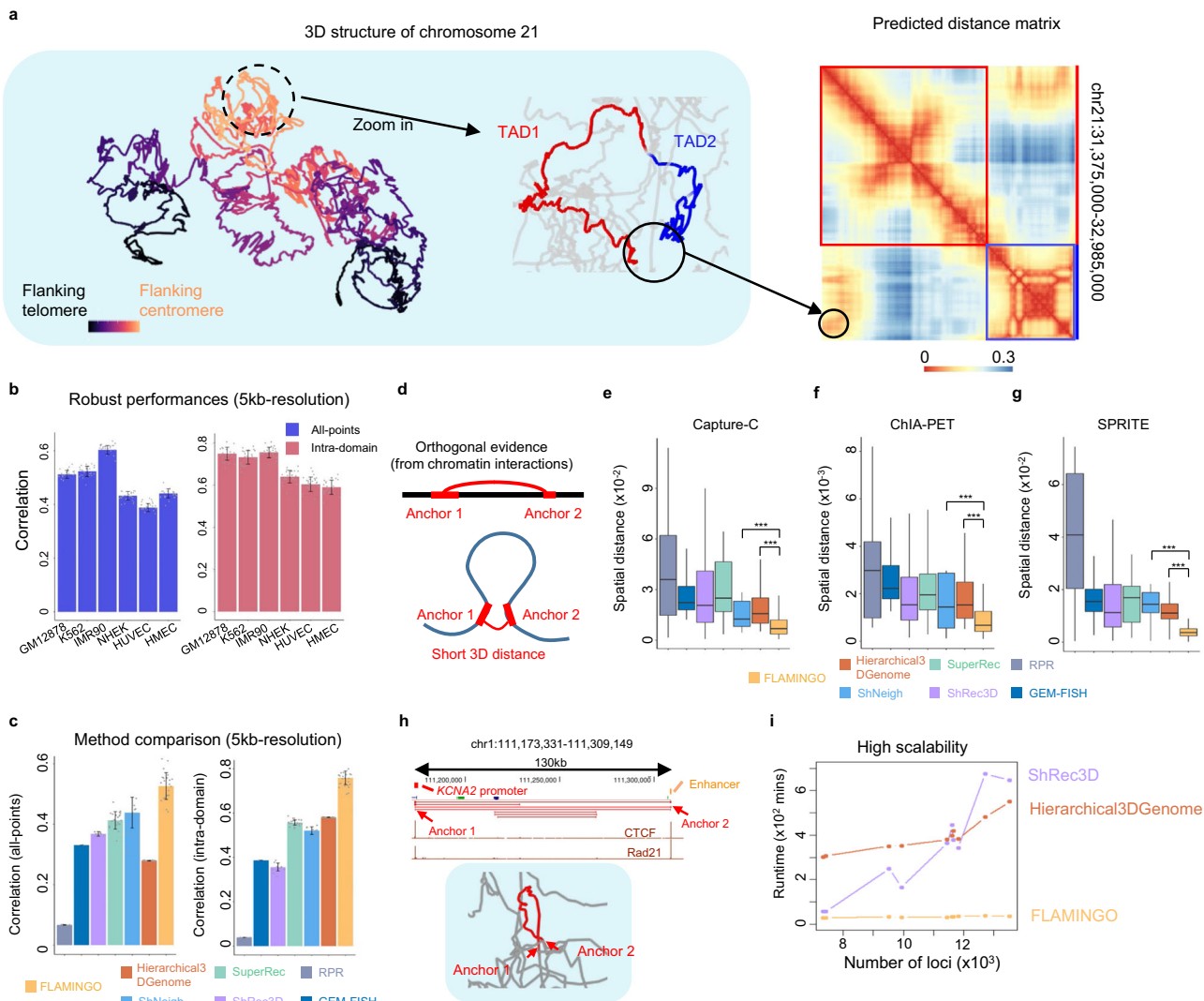

**Fig. 3 FLAMINGO demonstrates superior accuracy and scalability for high-resolution structure predictions. a** The reconstructed structure of chromosome 21 at 5 kb-resolution (left). The color gradient represents the genomic distance to the centromere (flanking centromere; yellow; flanking telomere: black). As an example, FLAMINGO recovers the chromatin loop formed by two TADs (chr21:31,375,000–32,985,000; middle), corresponding to inter-TAD hotspots in the reconstructed 3D distance matrix (right). **b** Robust performance of FLAMINGO across six cell-types at 5 kb-resolution. Correlations between predicted and observed distance matrices are calculated for all 5 kb fragments (all-points: blue) and fragments within domains (intra-domain: salmon). Error bars represent the standard deviations across $n = 23$ chromosomes. **c** Performance comparison with the state-of-the-art algorithms based on Hi-C data in GM12878 at 5 kb-resolution (all-points: left; intra-domain: right). Error bars represent the standard deviations across chromosomes with complete predictions ($n = 23$ for FLAMINGO, Hierarchical3Dgenome and SuperRec; $n = 10$ for ShRec3D; $n = 9$ for ShNeigh; $n = 6$ for RPR). GEM-FISH does not have error bars because it can only complete the prediction for chromosome 21. **d** Orthogonal chromatin interaction data provides additional evaluation metrics: anchors of chromatin interactions are expected to have short 3D distances. **e**–**g** FLAMINGO predicts significantly shorter distances between anchors of chromatin interactions profiled by Capture-C ($n = 3,692$) (**e**), ChIA-PET ($n = 214$) (**f**) and SPRITE ($n = 871$) (**g**). The statistical significance (***) is calculated by one-sided Mann–Whitney test: (**e**) p-value $= 9.4 \times 10^{-25}$ (orange) and p-value $= 7.6 \times 10^{-24}$ (blue); (**f**) p-value $= 2.8 \times 10^{-22}$ (orange) and p-value $= 5.1 \times 10^{-20}$ (blue); (**g**) p-value $= 7.4 \times 10^{-31}$ (orange) and p-value $= 6.5 \times 10^{-42}$ (blue). The 3D structures of different methods are normalized for fair comparison. The center lines of boxplots show the median, the upper and lower box limits show the 25th and 75th percentiles respectively. The whiskers extend up to 1.5 times the interquartile range away from the limits of the boxes. Outliers outside this range were removed from the figure. **h** One example of chromatin loops predicted by FLAMINGO for a significant ChIA-PET interaction (red links) linking the *KCNA2* promoter (red) with a distal enhancer (orange). **i** Comparison of the computational scalability by measuring the runtime (y-axis) as a function of different numbers of genomic loci (x-axis). Source data are provided as a Source Data file.

in chromosome 1, there are $6.7 \times 10^7$ pairs of 5 kb fragments with measured Hi-C contacts as constraints. Furthermore, the predicted intra-domain structures demonstrate higher correlations across the six cell-types (Fig. 3b, right), especially in GM12878, K562, and IMR90 (average correlation > 0.73). In addition, even at 1 kb-resolution, the reconstructed 3D structures achieve high correlations with the observed spatial distances for both whole

chromosomal structures and intra-domain structures (Supplementary Fig. 2, all-points correlations ~0.4 and intra-domain correlations ~0.6). These consistently high correlations indicate that FLAMINGO is able to capture both long-range genome folding patterns and detailed structures within domains.

FLAMINGO was then compared with other methods, GEM-FISH[34], ShRec3D[33], Hierarchical3DGenome[35], ShNeigh[38], RPR[36],

and SuperRec[37], which are state-of-the-art and recently developed algorithms representing different modeling strategies (Methods section and Supplementary Note 1). Strikingly, FLAMINGO achieved substantially higher correlations than all the other methods, for both whole chromosome structures and intra-domain structures, at 5 kb-resolution (Fig. 3c and Supplementary Figs. 11 and 12). For example, FLAMINGO achieved a correlation of 0.53 for whole chromosome structures in GM12878, while the other methods only achieved correlations below 0.45 (Fig. 3c, left). Similar advantage of FLAMINGO is also observed when the performance comparison is restricted to off-diagonal long-range inter-domain distances (Supplementary Fig. 11). Moreover, focusing on detailed intra-domain structures, FLAMINGO achieved a correlation of 0.76 in GM12878, while the other methods only achieved correlations below 0.6 (Fig. 3c, right). Similarly, FLAMINGO outperformed across all the other five cell-types at 5 kb-resolution (Supplementary Fig. 12).

To further leverage orthogonal data for performance comparisons, high-resolution chromatin interactions profiled by Capture-C[45], ChIA-PET[50], and SPRITE[46] experiments were used to evaluate whether the reconstructed structures assign short 3D distances between interacting anchors (Fig. 3d, Methods section). Remarkably, FLAMINGO consistently demonstrated higher accuracy than other methods across all three sets of experimental metrics (Fig. 3e–g). The reconstructed structures from FLA-MINGO assign statistically significant shorter 3D distances between anchors of chromatin interactions ($p$–value $< 2 \times 10^{-16}$, one-sided Mann–Whitney test), while other methods are less likely to capture the structural proximity for chromatin interactions. As an example (Fig. 3h), a long-range ChIA-PET interaction (~130 kb) on chromosome 1 links a distal enhancer element (the anchor 2) to the promoter region of gene *KCNA2* (the anchor 1), where both anchors are bound by CTCF and Rad21. Interestingly, in the reconstructed high-resolution structure by FLAMINGO, the enhancer and the promoter are in close proximity with each other and the genomic region in between forms a smooth chromatin loop. As comparison, the Hierarchical3DGenome algorithm does not assign a short spatial distance between the interacting enhancer and the *KCNA2* promoter (Supplementary Fig. 13a). Additional examples can be found in Supplementary Fig. 13b, c. These results not only provide rigorous evidence to validate the superior accuracy, but they also underscore the impacts of FLAMINGO on decoding the mechanisms underlying orchestrated gene regulation in 3D space.

**Advanced scalability for large-scale chromosome conformations**. High-resolution 3D structure modeling places stringent demands for performance, reliability, and more importantly, scalability on algorithms, since a large number of genomic loci and pairwise distances are used in the optimization procedure. Based on efficient information compression and matrix computation, the computational complexity of FLAMINGO is $O(kN^2)$, where $N$ is the number of genomic loci, such as the number of 5 kb DNA fragments, and $k$ is a small constant (Supplementary Note 1). For example, it only took 42 min and 2.2GB memory for FLAMINGO to reconstruct the 5 kb-resolution 3D structure for chromosome 1, the largest chromosome in the human genome (Methods, Supplementary Note 1). For chromosomes 2–22 and chromosome X, FLAMINGO was able to predict their structures even faster (Supplementary Fig. 14a). As comparison, the state-of-the-art algorithms all have inferior scalability. The running times for Hierarchical3Dgenome and ShRec3D increase rapidly when the number of genomic loci becomes large (Fig. 3i), while the other methods (i.e. SuperRec, ShNeigh, RPR, and GEM-FISH) are even slower (Supplementary Fig. 14b). Most of these methods

can only make predictions for short chromosomes (e.g. chr12–22) at 5 kb-resolution (Supplementary Note 1). Furthermore, because FLAMINGO can accurately predict the 3D structures based on a small subset of pairwise distances, the scalability of FLAMINGO can be improved further by down-sampling the distance matrix from Hi-C (Supplementary Fig. 14c). In addition, based on our tests of 1 kb-resolution reconstruction for all chromosomes in GM12878 (Supplementary Fig. 2), FLAMINGO can generate complete predictions for large chromosomes fast. For the largest chromosome (chr1), it takes <25 h using 200GB memory to reconstruct the 1 kb-resolution 3D structure. Therefore, FLA-MINGO provides drastic improvements on the computational scalability, which is much desired since a large number of Hi-C datasets are to be generated in the near future[51,52].

**Analysis of multi-way interactions and QTLs by FLAMINGO beyond 2D Hi-C contact maps**. To demonstrate the biological discoveries enabled by FLAMINGO that are not directly visible from 2D contact maps, the reconstructed 3D chromatin structures are used to resolve two important questions. First, we analyzed the predicted 3D structure's capability of capturing multi-way chromatin interactions. Spatially coordinated molecular processes frequently form multi-way interactions (e.g. 3-way, 4-way, or 5-way interactions) in 3D space[46,53,54], which play pivotal roles in coupled transcriptional and epigenetic activities[55]. However, Hi-C contact maps can only reveal pairwise 2-way chromatin interactions. Moreover, the high rates of missing data in Hi-C result in large genomic regions with almost no measured interactions, further limiting the capability of finding multi-way interactions from 2D contact maps. Since FLAMINGO recovers the whole spatial structure, we hypothesize that the predicted 3D structures can improve the identification of multi-way interactions. The multi-way chromatin interactions profiled by SPRITE experiments in GM12878[46] are used to justify this hypothesis. In addition to pairwise interacting anchors (Fig. 3g), the GM12878 structure predicted by FLAMINGO consistently assigns significantly shorter spatial distances among anchors of multi-way interactions in SPRITE (Fig. 4a, $p$-value $< 10^{-2}$, one-sided Wilcoxon test), compared to genomic-distance controlled random samples, suggesting the predicted 3D structures are in strong agreement with the higher-order organizations of multi-way interactions. More importantly, compared to using the Hi-C contact map derived distance matrix, the predicted 3D structure by FLAMINGO can capture more multi-way interactions (Fig. 4b, Supplementary Fig. 15a, b). Here, a multi-way interaction is considered to be captured if all interacting anchors are located in the same 3D spatial neighborhood, where all pairwise spatial distances between anchors are smaller than a specified threshold. As shown in Fig. 4b, across a wide range of thresholds on normalized spatial distances, FLAMINGO consistently demonstrates higher capabilities of discovering more 3-way interactions. Even if relaxed distance thresholds are used, 28.5% 3-way interactions from SPRITE experiments cannot be identified based on Hi-C contact map derived distance matrix, while being captured by FLAMINGO (Fig. 4b). It is because these 3-way interactions involve distal interacting anchors across very long-range genomic regions (median genomic distance = 2.32 Mb), where Hi-C contact maps suffer from high rates of missing data. Similar results are also found for 4-way and 5-way interactions (Supplementary Fig. 15a, b), where FLAMINGO achieves much higher advantages. Fig. 4c shows a representative example of a 3-way interaction that has been identified by SPRITE experiments[46]. The three interacting anchors are brought into spatial proximity based on the predicted loop structures, which are also highlighted in the predicted distance matrix (Fig. 4c, right). As comparison, the

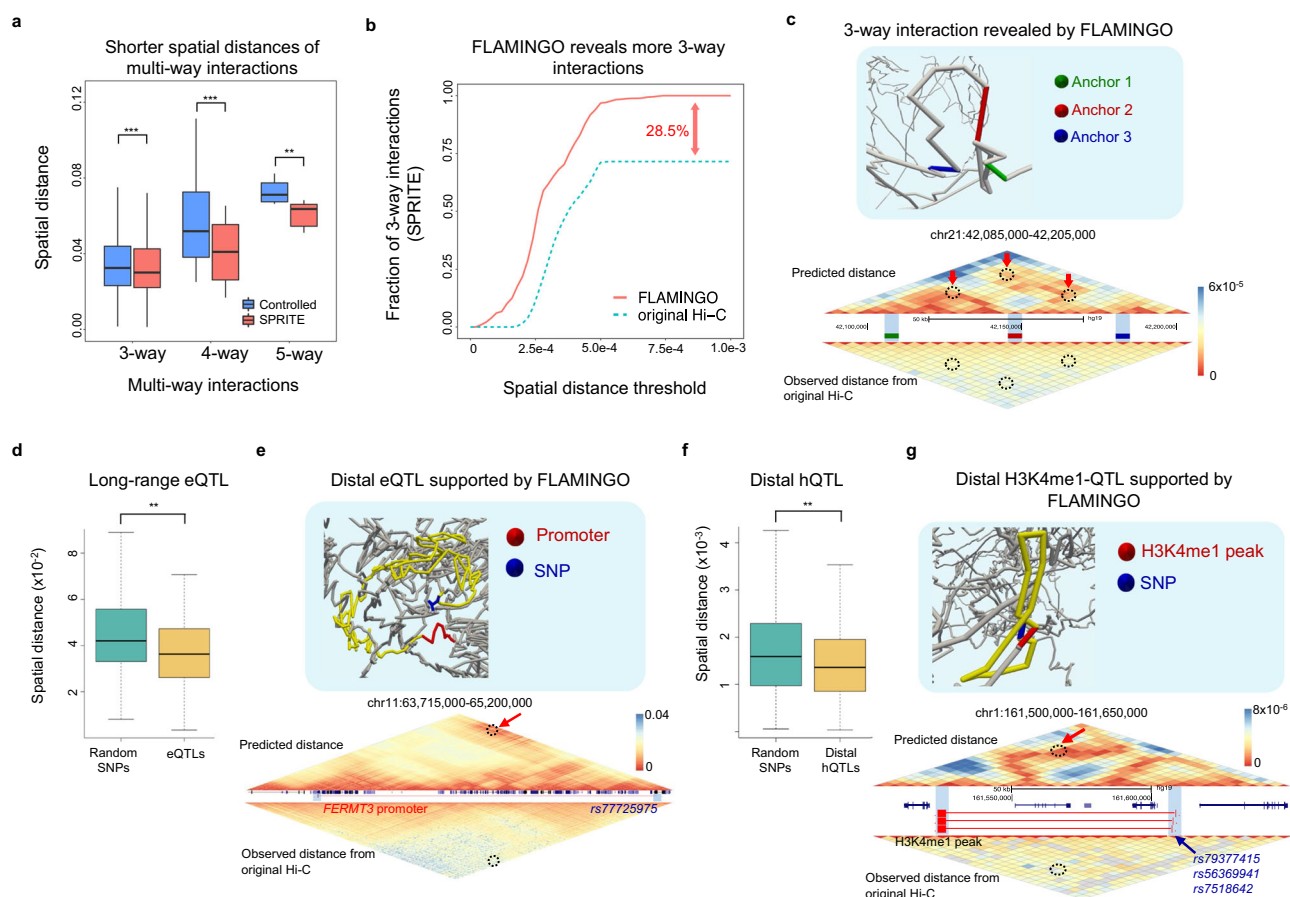

**Fig. 4 FLAMINGO interprets multi-way chromatin interactions and QTLs based on predicted 3D neighborhoods. a** SPRITE multi-way interactions on chr21 are predicted with shorter spatial distances than the genomic-distance controlled background (**$p$-value $< 10^{-2}$, ***$p$-value $< 10^{-3}$; one-sided Wilcoxon test). The $x$-axis corresponds to 3-way ($p$-value $= 2.3 \times 10^{-9}$, $n = 302$), 4-way ($p$-value $= 1.9 \times 10^{-4}$, $n = 17$), and 5-way interactions ($p$-value $= 7.8 \times 10^{-3}$, $n = 7$). The center lines of boxplots show the median, the upper and lower box limits show the 25th and 75th percentiles respectively. The whiskers extend up to 1.5 times the interquartile range away from the limits of the boxes. Outliers outside this range were removed from the figure. **b** FLAMINGO captures more 3-way interactions across different distance thresholds, compared to using normalized Hi-C contact map derived distance matrix. **c** One example of a SPRITE 3-way chromatin interaction captured by FLAMINGO. **d** The SNP-promoter pairs of long-range eQTLs (>900 kb) are assigned with significantly shorter spatial distances by FLAMINGO, compared to genomic-distance controlled random pairs (**$p$-value $= 4.1 \times 10^{-3}$ one-sided Wilcoxon test, $n = 1227$). The center lines of boxplots show the median, the upper and lower box limits show the 25th and 75th percentiles respectively. The whiskers extend up to 1.5 times the interquartile range away from the limits of the boxes. Outliers outside this range were removed from the figure. **e** One example of long-range eQTLs interpreted by FLAMINGO. The SNP rs77725975 (blue) and the promoter of *FERMT3* (red) are placed in close 3D proximity. **f** The SNP-H3K4me1 pairs of distal hQTLs are assigned with significantly shorter spatial distances by FLAMINGO, compared to genomic-distance controlled random pairs (**$p$-value $= 6.3 \times 10^{-3}$; one-sided Wilcoxon test, $n = 20,950$). The center lines of boxplots show the median, the upper and lower box limits show the 25th and 75th percentiles respectively. The whiskers extend up to 1.5 times the interquartile range away from the limits of the boxes. Outliers outside this range were removed from the figure. **g** One example of distal H3K4me1-QTLs interpreted by FLAMINGO. The SNPs rs79377415, rs56369941, rs7518642 (blue) and the H3K4me1 ChIP-seq peak (red) are placed in close 3D proximity. Source data are provided as a Source Data file.

distance matrix based on the Hi-C contact map shows no signals of spatial closeness for the three anchors. As another interesting example, a candidate 4-way interaction mediated by CTCF across a 12 Mb genomic region in chr1 is discovered by FLAMINGO, while the Hi-C based distance matrix shows no spatial patterns (Supplementary Fig. 15c). These results suggest that, by reconstructing 3D spatial structures, FLAMINGO can help to identify multi-way chromatin interactions and reveal higher-order genome organizations, beyond 2D Hi-C contact maps.

Second, we analyzed the predicted 3D structure's utility in interpreting genetic associations, such as long-range expression QTLs (eQTL) and distal histone QTLs (hQTL) in matched cell-types or tissues. QTLs statistically link genetic variants to molecular phenotypes and facilitate understandings of disease genetics. But it has been challenging to delineate the underlying molecular

mechanisms of genetic associations. Spatial proximity between genetic variants and target genes or histone modification peaks have been suggested to mediate genetic associations[56,57]. Similar to the approach of multi-way chromatin interaction analysis, the predicted 3D structure is evaluated with respect to its ability of interpreting QTLs[58–62] based on predicted short spatial distances, compared to using the Hi-C contact map derived distance matrix. Interestingly, across a wide range of thresholds on normalized spatial distances, substantially higher fractions of eQTLs and hQTLs are found to have their genetically associated loci (i.e. SNP-promoter or SNP-histone pairs) placed into small 3D neighborhoods by FLAMINGO (Supplementary Fig. 16). Focusing on the long-range eQTLs[61] whose SNPs and target gene promoters are >900 kb away, these SNP-promoter pairs are found to be assigned with significantly shorter spatial distances, compared to genomic-

distance controlled random pairs ($p$-value $= 1.3 \times 10^{-3}$, one-sided Wilcoxon test, Fig. 4d), suggesting the effectiveness of FLAMINGO in interpreting genetic associations. For each specific long-range eQTL ($>900$ kb), a random set of SNP-promoter pairs with the same genomic-distance from the same chromosome is generated (Methods). Among these long-range eQTLs ($n = 1227$), 671 of them (54.7%) are predicted to have spatial distances that are at least 2-fold shorter than the median spatial distances of genomic-distance controlled random pairs. As a representative example (Fig. 4e), the SNP rs77725975 is a significant long-range eQTL to the gene *FERMT3* ($p$-value $= 2.6 \times 10^{-4}$) in whole blood cells[61] with a genomic distance of 983 kb. This eQTL is placed into 3D proximity by FLAMINGO in GM12878, where the SNP rs77725975 and *FERMT3*'s promoter are located spatially close to each other, while the Hi-C based distance matrix fails to provide structural basis to interpret this eQTL. Similarly, distal hQTLs[62] are also found to be assigned with significantly shorter spatial distances by FLAMINGO, compared to genomic-distance controlled random pairs ($p$-value $= 2.84 \times 10^{-3}$, one-sided Wilcoxon test, Fig. 4f). Among the distal hQTLs ($n = 20,950$), 11,797 of them (56.3%) are predicted to have spatial distances that are at least 2-fold shorter than the median spatial distances of genomic-distance controlled random pairs. As shown in Fig. 4g for a set of distal hQTLs ($p$-value $< 1.8 \times 10^{-4}$), FLAMINGO reconstructs a loop structure which brings the SNPs close to the specific target H3K4me1 peak that is ~75 kb away. In contrast, the distance matrix derived from Hi-C contact maps shows no signal of long-range interactions in this region. These results strongly support the FLAMINGO's ability of interpreting the potential mechanisms of distal QTLs by leveraging the reconstructed spatial proximity information, a critical step further to decipher genetic associations with molecular phenotypes.

**Geometrical property of chromatin structures**. To gain additional insights into genome folding, 3D geometrical metrics are needed to describe the complex shapes of chromatin structures, which cannot be directly obtained from Hi-C contact maps. The reconstructed 3D structures provide a systematic platform for dissecting geometrical signatures of chromatin organization. To do this, we calculated the curvatures for every 5 kb genomic bin along the 3D curves of chromosomes. A larger curvature around a genomic region indicates the chromatin bends more sharply, while a smaller curvature suggests the region is relatively straight. Interestingly, the curvatures around TAD boundaries show significantly lower curvature than flanking genomic regions (Fig. 5a, $p$-value $= 2.2 \times 10^{-16}$, one-sided Mann–Whitney test). Considering the loop extrusion model[16], it suggests that, when a loop is established and the extrusion complex stops sliding, the DNA located around the extrusion complex is maintained rigid. In addition, genomic regions with large curvatures show significantly higher GC-contents (Fig. 5b), consistent with the increased flexibility of GC-rich DNA sequences[63,64] that may facilitate intra-TAD interactions.

**Reference structure to interpret single-cell variabilities**. Based on observations of recent single-cell Hi-C and imaging data[65–68], chromatin structure is dynamic and demonstrates variabilities across individual cells. The optimal consensus structure reconstructed from bulk tissue Hi-C by FLAMINGO thus provides a reference of chromatin folding aggregated from a pool of cells, which can be used as a basis to delineate and interpret the ensemble of chromatin configurations[69,70]. We compared FLAMINGO's predicted consensus structure to the single-cell structures profiled by diffraction-limited 3D imaging[68] to analyze their relationship. The image-based dataset[68] contains an ensemble of

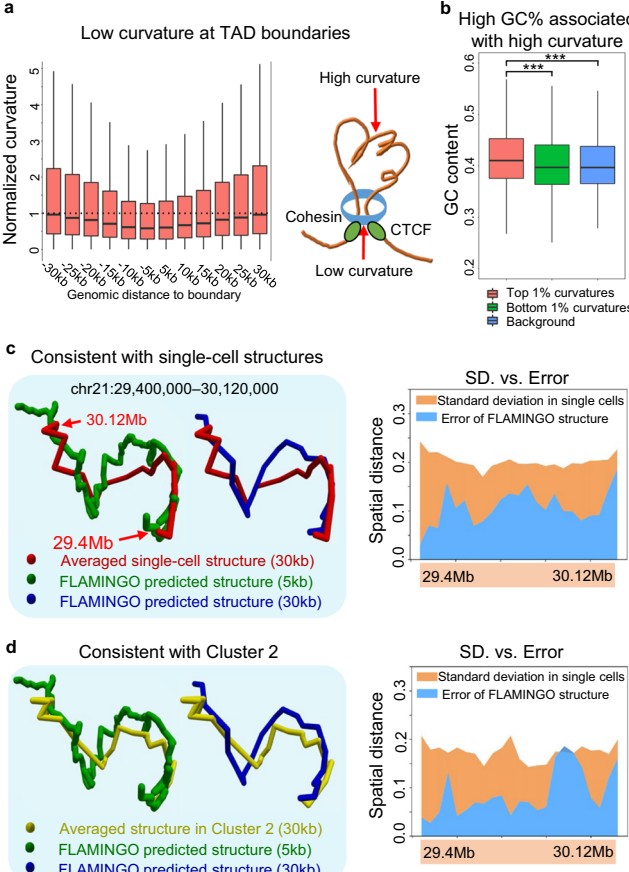

**Fig. 5 Geometrical signatures of predicted chromatin configuration and its comparison to single-cell structures. a** TAD boundaries demonstrate lower curvatures than flanking genomic regions. The center lines of boxplots ($n = 11,208$) show the median of normalized curvatures, the upper and lower box limits show the 25th and 75th percentiles respectively. The whiskers extend up to 1.5 times the interquartile range away from the limits of the boxes. Outliers outside this range were removed from the figure. **b** The regions with high curvatures show higher GC-content compared with genomic background. One-sided Mann–Whitney test ***$p$-value $= 2.7 \times 10^{-29}$ (green, $n = 5261$) and $p$-value $= 3.4 \times 10^{-34}$ (blue, $n = 5261$). The center lines of boxplots ($n = 5261$) show the median, the upper and lower box limits show the 25th and 75th percentiles respectively. The whiskers extend up to 1.5 times the interquartile range away from the limits of the boxes. Outliers outside this range were removed from the figure. **c** The consensus structure predicted by FLAMINGO consistently aligns with the average structure across single cells in K562. Right: the errors between the predicted consensus structure and the average structure (blue) are smaller than the intrinsic standard deviations among single cells (orange). **d** The consensus structure predicted by FLAMINGO is in strong agreement with the average structure across the subset of cells in cluster 2. Right: the errors between the predicted consensus structure and the cluster-2 specific average structure (blue) are smaller than the intrinsic standard deviations among single cells in cluster 2 (orange). Source data are provided as a Source Data file.

single-cell structures for a specific genomic region in chr21 at 30 kb-resolution. The averaged structure is calculated from the ensemble and is then compared with FLAMINGO's prediction. Fig. 5c shows the comparison for a loop structure in this region. Both 5 kb- and 30 kb-resolution predictions from FLAMINGO align well with the averaged structure of single cells (Fig. 5c, left). More importantly, the differences between these structures are

consistently smaller than the intrinsic standard deviations among single-cells within the ensemble (Fig. 5c, right), suggesting that the consensus structure can sufficiently quantify the major patterns of structural configurations. In addition, it suggests that the distance information derived from Hi-C contact frequency is overall consistent with the spatial configurations obtained from imaging techniques. To further analyze the structural variations relative to the consensus structure, the single-cell structures are classified into five different clusters, where individual cells belonging to the same clusters have similar structures. Structural variabilities are observed across distinct single-cell clusters. Interestingly, for the subset of cells in cluster 2, the cluster-specific average structure is highly similar to the predicted consensus structure (Fig. 5d, left), with the differences largely smaller than the intrinsic standard deviations among single cells within this cluster (Fig. 5d, right), further supporting the biological relevance of the predicted structure. The other four clusters also similarly demonstrate the overall folding patterns, each of which contains specific variations relative to the predicted consensus structure (Supplementary Fig. 17). Across all five clusters, the consensus structure consistently shows smaller differences to the cluster-specific average structures, than the intrinsic standard deviations of single cells within each cluster (Supplementary Fig. 17). These results suggest that the predicted consensus structures by FLAMINGO can facilitate improved interpretation of the structural heterogeneity in ensembles of single-cell structures.

**Robust performance to handle missing data in Hi-C datasets.** Due to limited sequencing depths of typical Hi-C experiments and low mappabilities of certain genomic regions, the observed distance matrices from Hi-C usually contain large portions of missing data[10,71], which present a very challenging problem for high-resolution modeling. For instance, considering the same Hi-C dataset for chromosome 1, the rate of missing data is 21% at 100 kb-resolution but quickly increases to 94.5% at 5 kb-resolution. Overall, the rate of missing data is >80% across chromosomes 1–22 and X in the human genome at 5 kb-resolution (Supplementary Fig. 14a). By incorporating the low-rank property of the distance matrix into the optimization procedure, FLAMINGO has the superior advantage of handling high rates of missing data.

To demonstrate FLAMINGO's capability of handling missing data, the observed distances derived from Hi-C were further down-sampled to check whether FLAMINGO still can reproduce the same high-resolution structures (Methods). As a representative example on chromosome 21 (chr21:34,000,000–35,000,000), FLAMINGO was able to robustly reconstruct the structure even if 50% of the observed pairwise distances from Hi-C was further down-sampled (Fig. 6a). By further down-sampling the dataset to the levels with only 20% and 5% of observed data remaining, FLAMINGO was still able to infer the loop structures formed by the four TADs in this region, with slightly increased intra-TAD fluctuations. In contrast, Hierarchical3DGenome predicted fuzzy structures with substantial fluctuations across all down-sampling rates. In addition, specific intra-TAD chromatin contacts were also captured by FLAMINGO, as shown by the specific hotspots within the TAD blocks in the predicted distance matrices at 50% and 70% of down-sampling rates (Fig. 6a), while Hierarchical3D-Genome only generated vague distance matrices without detailed structures within TAD blocks. More interestingly, FLAMINGO was also able to predict the short 3D distance for long-range inter-TAD contacts in the loop structure using only 70% of observed data (p-value = 0.038, permutation test, genomic distance controlled) (Fig. 6b), while Hierarchical3DGenome

predicted a much longer distance (p-value = 0.186). The predicted inter-TAD distance is in agreement with the original Hi-C distance matrix (Fig. 6c) and demonstrates a higher level of specificity, although it was inferred from down-sampled data. As additional justifications of the predicted structure with missing data (down-sampling rate = 70% or 50%), the specific intra- and inter-TAD chromatin contacts recovered by FLAMINGO, but not predicted by Hierarchical3DGenome, are supported by CTCF and cohesin bindings, along with convergent pairs of CTCF motifs (Fig. 6d, Supplementary Fig. 18a, and Supplementary Note 1).

As global quantitative evaluations, the recovered 3D structures and predicted distances by FLAMINGO using different down-sampled input matrices are compared with the originally observed distances. Strikingly, for the whole 5 kb-resolution distance matrix including both inter- and intra-domain structures, the correlation coefficients remain stable and high (~0.49), until <30% of observed distances from Hi-C are kept for predictions (Supplementary Fig. 18b). Focusing on detailed intra-domain structures, the correlation coefficients still remain to be robustly high (>0.74), until <50% of observed distances are kept (Fig. 6e). Across the wide range of down-sampling rates, FLAMINGO robustly achieves higher accuracy than other algorithms, based on comparisons using observed Hi-C contact maps (Fig. 6e and Supplementary Fig. 18b) and also other chromatin interaction datasets, such as Capture-C, ChIA-PET, and SPRITE (Supplementary Fig. 18c–e). For example (Fig. 6e), using only 10% of observed data, FLAMINGO achieved better accuracy than the state-of-the-art method, Hierarchical3DGenome, which used all of the observed data (Fig. 3c). These results clearly demonstrate FLAMINGO's ability to accurately reproduce high-resolution structures based on Hi-C with large fractions of missing data, which will significantly relax the demand of sequencing depths in Hi-C experiments and thus promote wide implementations of Hi-C in practice.

**Cross cell-type prediction of 3D structures.** Currently experimental Hi-C data have been collected only for a limited number of cell-types, due to the cost of the experiments or the difficulty of collecting sufficient numbers of cells for certain cell-types[71]. To enlarge the coverage of cell-types for 3D genome modeling, FLAMINGO is further extended to iFLAMINGO, an integrative version of the algorithm that can make cross cell-type predictions. To predict the 3D structure for a cell-type without Hi-C data, defined as target cell-type, iFLAMINGO combines two pieces of information (Fig. 7a and Methods section): (1) Hi-C data from another cell-type, defined as source cell-type, which provides the overall structural backbone of the genome; and (2) chromatin accessibility data, such as DNase-seq, from the target cell-type, which provides the cell-type-specific 1D epigenomic landscape. DNase-seq data are widely available across a large panel of cell-types and can characterize chromatin accessibilities at base pair resolution[6]. Since the levels of DNase-seq signals of a pair of genomic loci are associated with their 3D distances, for instance, co-accessible loci being significantly closer to each other in 3D space (Supplementary Fig. 19a), a regression model is built to impute approximate 3D distances based on DNase-seq signals in the target cell-type (Supplementary Fig. 19b). The imputed cell-type-specific distances are then incorporated into iFLAMINGO to predict the 3D genome structure in the target cell-type (Methods section).

iFLAMINGO was applied on the Hi-C data from GM12878 to predict the 3D genome structure in K562 by integrating K562-specific DNase-seq data into the modeling process. The resulting structure of chromosome 21 is shown in Fig. 7a

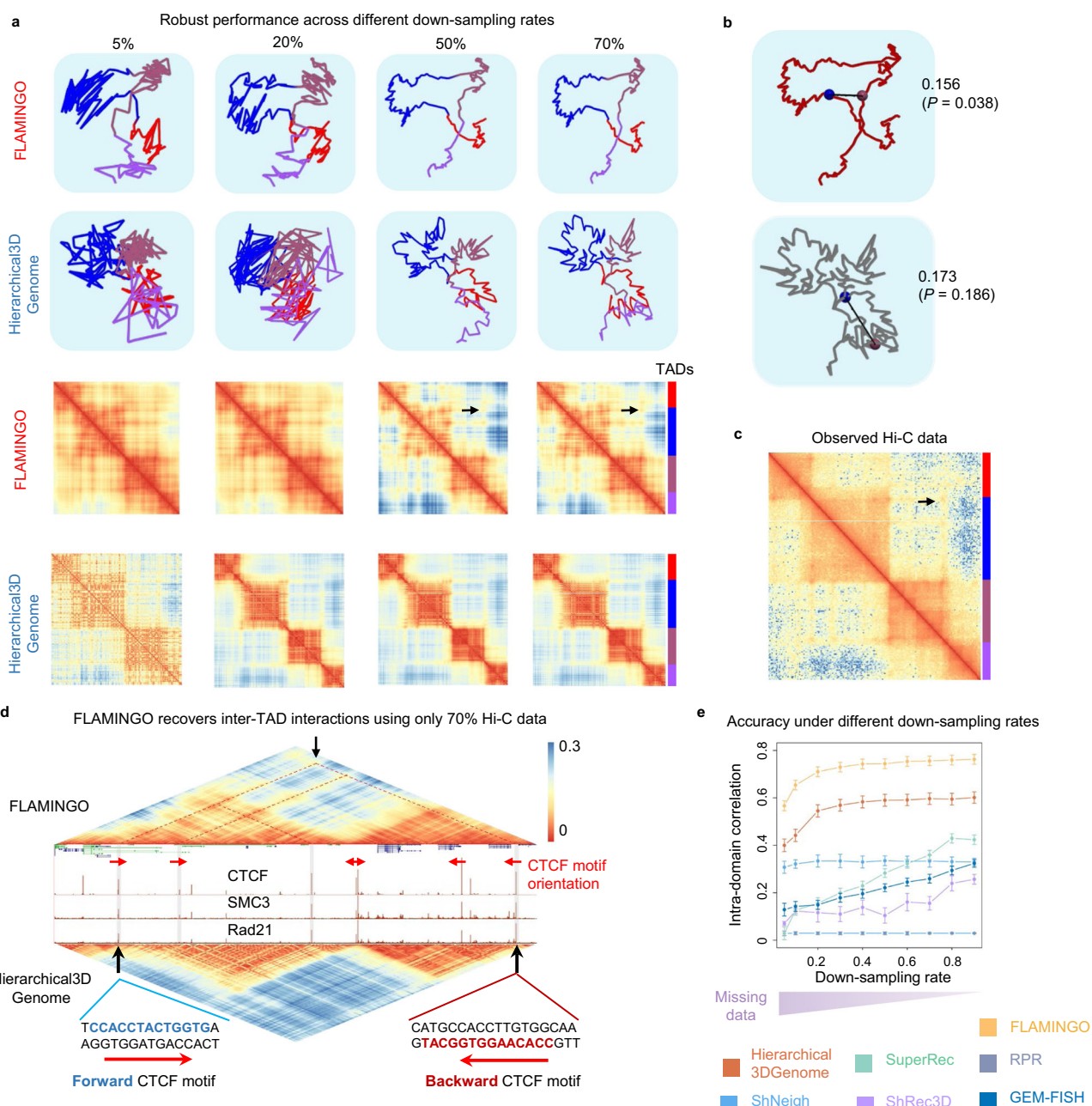

**Fig. 6 FLAMINGO robustly reconstructs 3D genome structures under different rates of missing data. a** Reconstructed 3D structures and completed distance matrices by FLAMINGO and Hierarchical3DGenome in chr21:34,000,000–35,000,000 using down-sampled data. As inputs, the observed distance matrix from Hi-C is down-sampled with different down-sampling rates (columns). Four TADs within this genomic region are annotated by colors. The inter-TAD interaction recovered only by FLAMINGO is highlighted by the black arrow. **b** FLAMINGO correctly recovers the short 3D distance between the two distal TAD boundaries (5′ of the blue TAD and 3′ of the brown TAD) as highlighted in **a**, with 70% down-sampled data. After normalization, FLAMINGO predicts a 3D distance of 0.156 ($p$-value $= 3.8 \times 10^{-2}$, $n = 1000$, permutation test, genomic distance controlled), while Hierarchical3DGenome predicts 0.173 ($p$-value $= 0.1862$, $n = 1000$, permutation test, genomic distance controlled). **c** The observed distance matrix from Hi-C data, along with TAD annotations and the highlighted inter-TAD interaction. **d** The inter-TAD interactions recovered by FLAMINGO (zoom-in view of the blue and brown TADs within chr21:34,100,000–34,850,000) are supported by CTCF and cohesin bindings and the convergent CTCF motifs (red arrows). The inter-TAD interactions are missed by Hierarchical3DGenome. **e** FLAMINGO achieves higher reconstruction accuracy against missing data. Correlations between predicted and observed intra-domain structures (the $y$-axis) are calculated for FLAMINGO and the state-of-the-art methods under different down-sampling rates (the $x$-axis). The dots show the average correlations based on $n = 10$ independently down-sampled input matrices and error bars correspond to the standard deviations across the ten random samples. Smaller down-sampling rates represent larger fractions of missing data. Source data are provided as a Source Data file.

(GM12878->K562). The 3D structure predicted based on GM12878 Hi-C alone is shown as the negative control, and the structure predicted directly from K562 Hi-C is included as the positive control (Fig. 7a). The GM12878->K562 structure not only captures the global structural signatures of the K562 genome but also reconstructs detailed loop structures more similar to K562, both of which are highlighted in Fig. 7a. By comparing with K562-specific chromatin interactions profiled by independent ChIA-PET experiments[72], the predicted 3D distances between interaction anchors from the

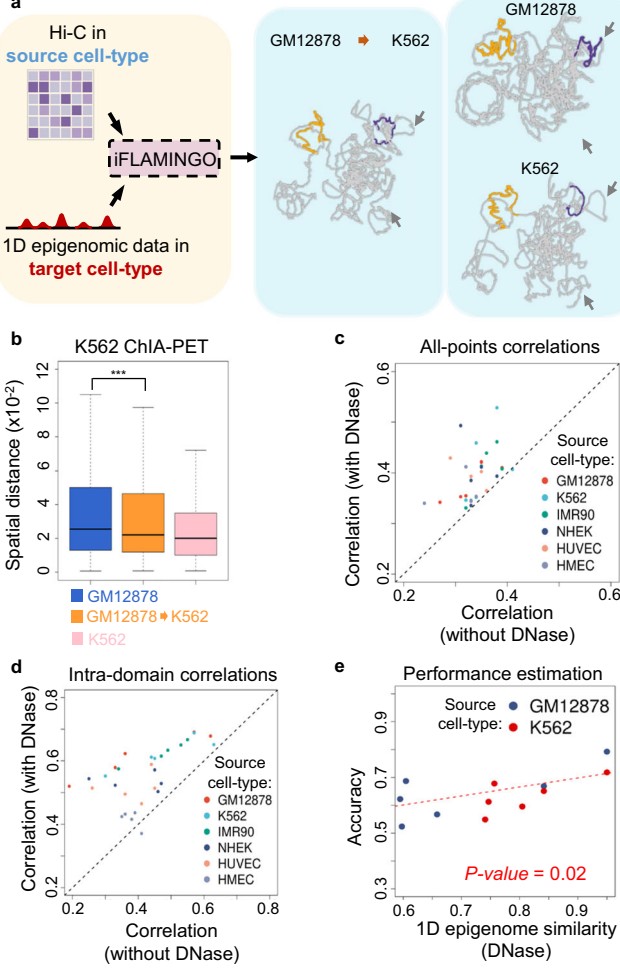

**Fig. 7 Cross cell-type predictions by iFLAMINGO. a** Hi-C data from the source cell-type and 1D epigenomics data from the target cell-type are integrated by iFLAMINGO to predict the 3D genome structure in the target cell-type (left). An example of the 3D structure of chromosome 21 for K562 predicted from GM12878 is shown (GM12878->K562). K562-specific structural properties are highlighted by arrows where iFLAMINGO correctly captures, while the GM12878-specific structure shows substantial differences. Two intra-domain structures are further highlighted in the three structures (orange and purple). **b** Comparison of 3D distances between interacting ChIA-PET anchors based on the predicted 3D structures of GM12878 (blue), GM12878->K562 (orange) and K562 (pink). $p$-value = $3.0 \times 10^{-4}$ ($n = 1562$, one-sided Mann–Whitney test). The center lines of boxplots show the median, the upper and lower box limits show the 25th and 75th percentiles respectively. The whiskers extend up to 1.5 times the interquartile range away from the limits of the boxes. Outliers outside this range were removed from the figure. **c, d** Performance comparisons between iFLAMINGO (the $y$-axis) and FLAMINGO (the $x$-axis) on cross cell-type predictions for $n = 30$ source-target pairs. Source-target pairs are colored by source cell-types. The performance is quantified by correlations between predicted and observed distances for all DNA fragments, i.e. all-points, in **c** and fragments within the same domains, i.e. intra-domain, in **d**. **e** Performance estimation (correlation of 3D distances, $y$-axis) for cross cell-type predictions of intra-domain structures as a function of 1D epigenomic similarities between cell-types (correlations of genome-wide DNase-seq data, $x$-axis). The regression line is fitted based on cross cell-type predictions from GM12878 and K562 ($p$-value = 0.02, $n = 12$, two-sided Student's $t$-test). Source data are provided as a Source Data file.

GM12878->K562 structure are significantly shorter than the distances from the GM12878 structure (Fig. 7b, $p$-value = 0.0003, one-sided Mann–Whitney test), suggesting the quantitatively improved similarity between the GM12878->K562 and K562 structures. Furthermore, the predicted spatial distances in the GM12878->K562 structure achieve a higher correlation with the experimentally-derived spatial distances of K562 Hi-C (correlation = 0.62, Supplementary Fig. 19c), compared to the correlation achieved by the basic experimentally-derived spatial distances of GM12878 Hi-C with the experimentally-derived spatial distances of K562 Hi-C (correlation = 0.55), suggesting that the predicted GM12878->K562 structure by iFLAMINGO captures the cell-type specificity of K562.

iFLAMINGO was further applied on all source-target pairs from the six cell-types with Hi-C data, and the performance was evaluated based on the correlations between predicted and observed distance matrices in target cell-types (Methods section). As comparison, the optimal structures predicted by FLAMINGO without using DNase-seq data are included as negative controls. Among all the 30 source-target cell-type pairs, iFLAMINGO achieved a higher accuracy for almost all the cross cell-type predictions (Supplementary Fig. 20), not only for the whole distance matrices (Fig. 7c) but also for intra-domain structures (Fig. 7d). These consistent improvements underscore iFLAMINGO's ability of cross cell-type structure predictions and highlight the importance of 1D epigenomic information in 3D genome modeling.

To further demonstrate iFLAMINGO's potential on enlarging the cell-type coverage for 3D structure reconstructions, the accuracy of cross cell-type 3D predictions is plotted as a function of 1D epigenomic similarities between the source and target cell-types (Fig. 7e). Using GM12878 or K562 as source cell-types, the accuracy of predicted intra-domain 3D structures in target cell-types is significantly associated with the 1D epigenomic correlations to the source cell-types ($p$-value = 0.02). Based on the fitted linear function, to obtain a cross cell-type prediction with accuracy>0.6, which is a level already higher than the state-of-the-art methods using Hi-C directly from the target cell-types (Fig. 3c), iFLAMINGO only requires Hi-C data available from a source cell-type with medium 1D epigenomic similarities (correlation > 0.65). Combined with the ongoing experimental efforts of chromatin characterizations, such as the 4D Nucleome Consortium[51], iFLAMINGO will substantially expand the catalog of cell-types with high-resolution 3D structures.

**Boost the resolution of 3D structures from low-resolution Hi-C.** Since another limiting factor of experimental Hi-C data is the resolution of contact maps being low[73,74], a tradeoff of genome-wide coverage of sequencing reads, it is much desired to predict high-resolution 3D structures from low-resolution contact maps of Hi-C. By incorporating high-resolution 1D epigenomic data, such as DNase-seq, iFLAMINGO is able to boost the resolution of the predicted 3D genome structures (Fig. 8a and Supplementary Fig. 19). After splitting low-resolution DNA fragments into high-resolution bins, DNase-seq signals help delineate the distance ambiguity across consecutive bins and fine-tune the structures through optimization (Methods section).

As a representative example, FLAMINGO was applied to the 25 kb-resolution distance matrix for chromosome 10, resulting in a 25 kb-resolution 3D structure (Fig. 8b, left). On the other hand, based on the 5 kb-resolution distance matrix, the 5 kb-resolution structure was generated by FLAMINGO as the benchmark

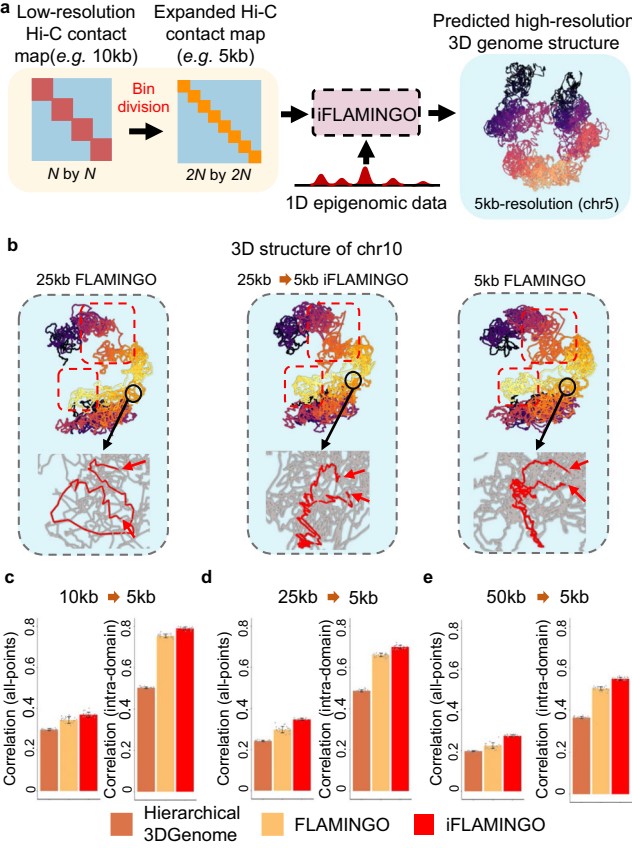

**Fig. 8 iFLAMINGO predicts high-resolution 3D structures from low-resolution Hi-C matrix by integrating 1D epigenomics datasets. a** Scheme of the high-resolution 3D structure prediction. Low-resolution distance matrix from Hi-C for *N* large DNA fragments of size 10 kb, are divided into smaller DNA fragments of size 5 kb, resulting in a 2*N* by 2*N* distance matrix, where the small DNA fragments inherit the same distances to other fragments from the original large fragment. The high-resolution 1D epigenomics signals in each small DNA fragment are integrated into iFLAMINGO to predict the high-resolution 3D genome structures. As one example, the 3D structure of chromosome 5 at 5 kb-resolution predicted from the 10 kb-resolution distance matrix is shown. **b** Example of the predicted 5 kb-resolution 3D structure of chromosome 10 from 25 kb-resolution distance matrix (middle, 25 kb->5 kb), compared with the 25 kb-resolution structure (left) and the 5 kb-resolution structure (right). The large-scale structural differences are highlighted by red boxes. The comparisons of detailed intra-domain structures (red) are shown in inset. The red arrows represent the boundaries. **c–e** Performance comparison of predicting 5 kb-resolution structures from 10 kb-resolution (**c**), 25 kb-resolution (**d**), and 50 kb-resolution distance matrices (**e**). Correlations between predicted and observed 5 kb-resolution distances are calculated for all DNA fragments, i.e. all-points, and for fragments within the same domains, i.e. intra-domain. The bar plot shows the average correlations across *n* = 23 chromosomes and the error bars show the standard deviations across 23 chromosomes. Data are presented as mean values ±SD. Source data are provided as a Source Data file.

structure (Fig. 8b, right). Finally, applying iFLAMINGO on the 25 kb-resolution distance matrix, along with the DNase-seq data, led to a 5 kb-resolution structure, the 25 kb->5 kb structure (Fig. 8b, middle), which shows increased similarity to the 5 kb-resolution benchmark structure. The 25 kb->5 kb structure not only captures large-scale structural properties but also recovers the detailed high-resolution loops in the 5 kb-resolution structure, which are missing in the 25 kb-resolution structure (Fig. 8b).

To quantitatively evaluate the accuracy of boosted resolution genome-wide, a series of low-resolution distance matrices, at 10 kb, 25 kb, and 50 kb resolution, respectively, were generated from the same Hi-C datasets. The reconstructed structures were then compared with the original 5 kb-resolution distance matrix. Across all the tests, iFLAMINGO achieved the highest correlations to the benchmark structures (Fig. 8c–e). For instance, using 10 kb-resolution distance matrices as inputs, iFLAMINGO achieved an average correlation of 0.37 for the whole reconstructed 5 kb-resolution matrices and an average correlation of 0.79 for intra-domain matrices, both of which are higher than the state-of-the-art methods even when they were directly applied on 5 kb-resolution input matrices (Fig. 3c). Therefore, iFLAMINGO not only substantially improves the information extraction from low-resolution Hi-C data but will also widely facilitate the implementation of Hi-C protocols without stringent constraints on resolution.

## Discussion

In this study, we have developed an algorithm, FLAMINGO, to reconstruct high-resolution spatial conformations for large genomes in 3D space. Using low-rank matrix completion techniques, FLAMINGO is able to substantially improve data mining efficiency for Hi-C experiments. Based on a series of rigorous performance evaluations, FLAMINGO consistently demonstrates superior accuracy and advanced scalability compared to other state-of-the-art methods. The strong agreements between the predicted genome architectures and orthogonal experimental evidence, such as Capture-C, ChIA-PET, and SPRITE, further highlight FLAMINGO's ability of capturing high-resolution spatial signatures of chromatin. Biologically, the reconstructed 3D structures facilitate additional discoveries and understandings, beyond 2D contact maps, such as higher efficiency of identifying multi-way chromatin interactions, interpretation of long-range QTLs, geometrical properties associated with TAD boundaries, and providing structural references to analyze single-cell variabilities of chromatin folding. Furthermore, FLAMINGO, along with its integrative version iFLAMINGO, addresses four fundamental challenges in 3D genome modeling: (1) high scalability to reconstruct high-resolution 3D structures for all chromosomes from massive Hi-C datasets; (2) robust performance to handle large portions of missing data in Hi-C; (3) accurate cross cell-type prediction of 3D structures for cell-types lacking Hi-C datasets; and (4) boosting the resolution of reconstructed 3D structures from low-resolution Hi-C contact maps. Given all these advantages, FLAMINGO will be an important tool for both computational and experimental studies on 3D genomes. The reconstructed high-resolution structures across different cell-types will significantly facilitate biological insights into the spatial organization of chromatin and its underlying mechanisms.

As one of the major benefits of FLAMINGO, the generated high-resolution 3D structures can serve as a platform to understand how transcriptional regulation is modulated in 3D space. Overlaid with functional genomics data, FLAMINGO predictions provide high-resolution structural supports for long-range regulatory links between enhancers and promoters (Fig. 3e–h), and recover the short 3D distances between CTCF-associated boundaries of chromatin loops (Fig. 6a–d and Supplementary Fig. 1). Moreover, beyond 2D chromatin contact maps, FLAMINGO can help to analyze higher-order multi-way interactions (Fig. 4a–c) and long-range cis-regulatory QTLs (Fig. 4d–g), and characterize geometrical signatures of chromatin shapes (Fig. 5a–b). In recent years, deep learning models have been developed to predict regulatory interactions in gene regulation and TAD organization from DNA sequences[75,76]. Since

FLAMINGO and deep learning models have complementary algorithmic strengths, it is expected to gain system-level knowledge on the relationship between gene regulation and chromatin organization by combining FLAMINGO with these deep learning algorithms.

The optimized consensus structure provides an efficient representation of the 3D genome for biologists with the advantage of high interpretability. Another type of methods aim to infer variations of the underlying chromatin structures, namely ensemble structures, using either polymer simulation models[77–79] or machine learning algorithms[69,70]. While modeling structural variations is important, it is sometimes difficult to biologically interpret an individual structure from a pool of predictions and to delineate experimental cell-to-cell variations from the increased noisy fluctuations. As shown in the comparisons between the reconstructed structure and the ensemble of single-cell structures, including both ensemble average structures and variable cluster-specific structures (Fig. 5c, d), FLAMINGO's predictions can serve as effective reference structures to standardize the relative variabilities across single cells. Equipped with the complementary advantages of accuracy and robustness against noise, FLAMINGO can help the ensemble-structure learning algorithms to improve both the predictive performance and the interpretation of structures.

There are currently two limitations of FLAMINGO, which require future methodology developments. First, although the transformation function from Hi-C contact frequency to spatial distance has been justified for intra-chromosomal contacts by previous studies[14,34] and our analyses (Fig. 5c–d, Supplementary Fig. 4, and Supplementary Fig. 17), there is currently no systematic estimation of the function for inter-chromosomal contacts. Thus, FLAMINGO can only reconstruct 3D structures for each chromosome separately, while it is difficult to assemble the structure for the whole genome including inter-chromosomal distances. Similarly, due to the lack of sequencing reads, centromere and telomere regions are excluded from the reconstruction of spatial chromosome conformations. These regions, especially centromere regions that have been demonstrated to be important in regulating chromatin organization by previous studies[69,80], are components that should not be excluded if organizations for the whole genome are to be assembled. In order to achieve complete reconstructions of 3D genome, future algorithmic developments will be needed to overcome this limitation. Second, the consensus structure predicted by FLAMINGO represents the population-average architecture from large numbers of cells, which can not capture the highly dynamic property of 3D chromatin[81,82] (such as the dynamic chromatin loops and TADs). The multi-scale spatial conformation of chromosomes varies from cell to cell[83] and the variability plays important roles in epigenetics, gene regulation, and DNA damage repair[84]. A series of ensemble-structure prediction algorithms have been developed to explore the dynamic conformations[69,70,77–79]. As a future development that can help to further overcome this limitation, single-cell Hi-C datasets will be needed to predict 3D structures for individual cells. Single-cell Hi-C datasets are highly sparse and raise significant challenges in handling missing data. Although FLAMINGO demonstrates superior performance against missing data for bulk tissue Hi-C datasets even with ~98% missing rate at 5 kb-resolution (Fig. 6e, corresponding to 50% down-sampling rate), typical single-cell Hi-C experiments have >99.99% missing rates at 100 kb-resolution. Therefore, the highly sparse single-cell Hi-C datasets require further algorithmic improvements, in order to characterize the detailed structural variations across individual cells.

Overall, the combined strengths of handling large rates of missing data, making cross cell-type predictions, and boosting resolutions, suggest high impacts of FLAMINGO on 3D genome analyses. High-resolution structures can be inferred for diverse panels of cell-types spanning different differentiation lineages, without increasing sequencing depths or requiring closely similar cell-types. Thus, it will not only improve the data mining of existing Hi-C data but also address the urgent need from large-scale Hi-C data resources to be generated in the near future, such as the 4D Nucleome Consortium. Together with the recent image-based 3D genome information[4] and the high-dimensional epigenomics data[6,85], FLAMINGO is expected to substantially expand our understandings of the spatially orchestrated genome architectures across cell-types.

## Methods

**Chromatin contact maps and epigenomics datasets.** We collected the Hi-C chromatin contact maps of six human cell-types, including GM12878, K562, IMR90, HMEC, HUVEC, and NHEK, from the GEO database[10] (GEO:GSE63525). To remove potential biases in the Hi-C data, we normalized chromatin interaction-frequency matrices using the Knight-Ruiz normalization method as suggested by previous studies[10]. The normalized Hi-C interaction frequencies are then transformed into 3D Euclidean distances based on the exponential function[14]: $D_{ij} = IF_{ij}^{(-\eta)}$, where $D_{ij}$ represents the squared pairwise 3D distance between DNA fragments $i$ and $j$, $IF$ represents the interaction frequency, and $\eta$ is a free parameter. In fact, after testing our model by taking different values of $\eta$ in the range suggested by previous experimental estimates[14,28], we have found that the accuracy of reconstruction is robust to the choice of $\eta$ (Supplementary Fig. 4). Therefore, by default, $\eta$ is set to 0.5 ($\eta/2 = 0.25$) as suggested by previous literature[14]. The validity of 3D distances converted from Hi-C contact maps, which are termed as observed distances from Hi-C in this paper, are also supported by the high similarity between the reconstructed structure and averaged structures of single-cell clusters, whose 3D configurations are directly obtained from imaging data (Fig. 5c, d and Supplementary Fig. 17).

The genome-wide DNase-seq datasets of chromatin accessibility from the six cell-types were collected from the ENCODE and Roadmap consortia[50,86]. In a specific cell-type, for each DNA fragment, the averaged DNase-seq signal (namely fold-change over genomic background) within the fragment is used to represent the cell-type specific chromatin accessibility in the genomic locus. Additional details on data collection and preprocessing are given in Supplementary Note 1.

**Model framework of FLAMINGO.** FLAMINGO reconstructs 3D genome structures based on Hi-C chromatin contact maps using the low-rank matrix completion technique (Fig. 1a), which can efficiently delineate underlying low-rank structures from the large and noisy pairwise distance matrices. The cell-type specific 3D coordinates of high-resolution DNA fragments for each chromosome are predicted by solving a constrained rank-minimization problem using the augmented Lagrangian method[48], which can converge fast and can robustly handle large amounts of missing data.

To enable parallel computation, a hierarchy of two scales (1 Mb and 5 kb) is used to model each chromosome and an integrative assembly strategy is designed to build optimal high-resolution chromosomal structures from these two scales (Supplementary Fig. 3). Based on simulated benchmark analysis, the performance of FLAMINGO does not rely on specific choices of resolutions or domain partitions (Supplementary Fig. 7). In addition, an integrative variant of FLAMINGO, iFLAMINGO (Fig. 7a and Supplementary Fig. 19), is also developed to incorporate cell-type-specific DNase-seq datasets into the model so as to (1) enable cross cell-type predictions and (2) boost resolution of predicted 3D genome structures.

**Reconstruct 3D genome structures based on low-rank matrix completion.** Each chromosome is modeled as a 'beads-on-a-string' polymer chain, where each DNA fragment is modeled as a bead, and the centromere and telomere regions are removed from the analysis as suggested by previous studies[33–35]. Structure reconstruction requires inferring the optimal 3D coordinates of consecutive DNA fragments along a chromosome, which maximally align with the pairwise 3D distances between DNA fragments observed from Hi-C data. A key property of FLAMINGO is its capability to leverage the low-rank nature of a pairwise distance matrix from Hi-C; namely, the high-dimensional pairwise distance matrix is biologically generated by the underlying low-rank coordinate matrix of DNA fragments (rank ≤ 3). Defined by the coordinate matrix ($\mathbf{P}$), the Gram matrix ($\mathbf{X} = \mathbf{PP}^T$) has a rank ≤3. The squared Euclidean distance matrix ($\mathbf{D}$) is a sum of three matrices: $\mathbf{D} = \text{diag}(\mathbf{X})\mathbf{1}^T + \mathbf{1}^T \text{diag}(\mathbf{X}) - 2\mathbf{X}$ where rank$(\mathbf{X}) \leq 3$, rank $(\text{diag}(\mathbf{X})\mathbf{1}^T) \leq 1$, and rank$(\mathbf{1}^T\text{diag}(\mathbf{X})) \leq 1$. Due to the property of ranks for matrix addition, the Euclidean distance matrix has a rank ≤ 5. Based on the theory of matrix completion[42], the low-rank property of both the pairwise Euclidean distance matrix (rank ≤ 5) and the Gram matrix (rank ≤ 3) guarantees that, under certain randomness assumptions on measurements, the underlying 3D structure can be predicted using a small fraction of data from Hi-C (Fig. 1a).

We define $\mathbf{P}$ as the $N$ by 3 coordinate matrix for $N$ consecutive DNA fragments along a chromosome. We also define $D_{i,j}$ as the squared 3D spatial distance between DNA fragments $i$ and $j$. Thus, the objective function for 3D genome reconstruction is:

$$\min ||\mathbf{X}||_*$$

$$\text{subject to } X_{i,i} + X_{j,j} - 2X_{i,j} = D_{i,j}, (i,j) \in \Omega; \mathbf{X1} = 0; \mathbf{X} = \mathbf{X}^T; \mathbf{X} \text{ is positive semidefinite}$$
(1)

where $\mathbf{X} = \mathbf{PP}^T$ is the Gram matrix, $||\mathbf{X}||_*$ represents the nuclear norm $\text{Tr}(\sqrt{\mathbf{X}^T\mathbf{X}})$, which is related to the rank of matrix $\mathbf{X}$, and the measurement set $\Omega$ represents a subset of indices of DNA fragment pairs. We further introduce a linear sampling operator $A$ as:

$$A(\mathbf{X}) = f \in R^{|\Omega|*1}, f_i = <\mathbf{X}, \boldsymbol{\omega}_{\alpha_i}> \text{ for } \alpha_i \in \Omega$$
(2)

where $\alpha_i = (\alpha_{i,1}, \alpha_{i,2})$ is the index of a DNA fragment pair. The matrix basis $\boldsymbol{\omega}_{\alpha_i}$ is defined as:

$$\boldsymbol{\omega}_{\alpha_i} = \mathbf{e}_{\alpha_{i,1},\alpha_{i,1}} + \mathbf{e}_{\alpha_{i,2},\alpha_{i,2}} - \mathbf{e}_{\alpha_{i,1},\alpha_{i,2}} - \mathbf{e}_{\alpha_{i,2},\alpha_{i,1}}$$
(3)

where $\mathbf{e}_{i,j}$ represents a matrix which has 1 at entry $(i,j)$ and 0 otherwise. For later use, we define the adjoint of $A$ as $A^*$, where $A^*\mathbf{y} = \sum_i y_i \boldsymbol{\omega}_{\alpha_i}$. The subset of DNA fragment pairs ($\Omega$ and $\alpha_i$) is randomly down-sampled from all measured pairs of DNA fragments with specified down-sampling rates. Intuitively, by defining $\omega$ and $\alpha_i$, the linear operator $A$ summarizes all the constraints in one notation so that the objective function can be re-written in a compact form:

$$\min_{\mathbf{P}} \text{Trace}(\mathbf{PP}^T), \text{ subject to } A(\mathbf{PP}^T) = \mathbf{b}$$
(4)

where $\mathbf{b} = A(\mathbf{M})$ and $\mathbf{M}$ represents the true underlying low-rank Gram matrix from Hi-C data satisfying $M_{i,i} + M_{j,j} - 2M_{i,j} = D_{i,j}$.

A penalization term is further added to the objective function to control unexpected large distances predicted between adjacent DNA fragments caused by low Hi-C data quality at certain genomic locations. Therefore, the final objective function is:

$$\min_{\mathbf{P}} \text{Trace}(\mathbf{PP}^T) + \lambda/2 ||B(\mathbf{PP}^T) - d^t\mathbf{1}||_2^2, \text{ subject to } A(\mathbf{PP}^T) = \mathbf{b}$$
(5)

where $\lambda$ represents the penalization parameter, and the scalar $d^t$ represents the maximal allowed distance between adjacent DNA fragments. The linear measurement operator $B$ projects the Gram matrix to the sub-diagonal elements:

$$B(\mathbf{X}) = g(\mathbf{X}) \in R^{(n-1)*1}, \text{ where } g_i(\mathbf{X}) = <\mathbf{X}, \boldsymbol{\omega}_{\beta_i}> \text{ for } \beta_i = (i, i+1), \text{ and } \mathbf{1} \in R^{(n-1)*1}.$$
(6)

The adjoint of $B$ is denoted as $B^*$, where $B^*\mathbf{y} = \sum_i y_i \boldsymbol{\omega}_{\beta_i}$.

Intuitively, the low-rank matrix completion model only needs a subset of the whole set of pairwise distances, which is indexed by $\Omega$, to reconstruct the Gram matrix $\mathbf{PP}^T$, and it requires the optimal matrix $\mathbf{PP}^T$ to follow three properties (Fig. 1a): (1) The rank of matrix $\mathbf{PP}^T$ should be as small as possible by minimizing the trace of $\mathbf{PP}^T$. This property is consistent with the low-rank assumption for 3D chromatin structures; (2) The pairwise distances based on the reconstructed 3D coordinates of DNA fragments should align with the subset of 3D distances indexed by $\Omega$ by satisfying the optimization constraints. This ensures that the model can accurately reconstruct 3D genome structures consistent with observed pairwise distances; (3) The 3D distances between adjacent DNA fragments are bounded. This constraint removes unrealistically stretched structures of chromatin and guarantees a smooth genome structure.

Since the trace function $\text{Trace}(\mathbf{PP}^T)$ is convex with respect to $\mathbf{P}$, we solve the optimization problem by the alternating-direction method of multipliers[49]. The augmented Lagrangian is given by:

$$L(\mathbf{P};\Lambda) = \text{Trace}(\mathbf{PP}^T) + \lambda/2||B(\mathbf{PP}^T) - d^t\mathbf{1}||_2^2 + r/2||A(\mathbf{PP}^T) - \mathbf{b} + \Lambda||_2^2$$
(7)

where $\lambda$ is the penalty parameter, $r$ is the regularization parameter, and $\Lambda$ is the Lagrangian multiplier. The gradient of the augmented Lagrangian with respect to $\mathbf{P}$ is given by:

$$2\mathbf{P} + 2\lambda B^*(B(\mathbf{PP}^T) - d^t\mathbf{1})\mathbf{P} + 2rA^*(A(\mathbf{PP}^T) - \mathbf{b} + \Lambda)\mathbf{P}.$$
(8)

Starting from $\Lambda = 0$ and a random initial guess for $\mathbf{P}$, the following iteration will continue until the error between the reconstructed and observed distances indexed by $\Omega$ is smaller than a specified threshold (default=$10^{-3}$): $\mathbf{P}$ is updated with the Barzilai-Borwein steepest descent method using the current $\Lambda$ and then $\Lambda$ is updated using the current $\mathbf{P}$[49]. The accuracy of the model does not rely on the value of $r$ and $\lambda$, and we have set the parameters $r = 1$ and $\lambda = 10$ based on the previous study of low-rank reconstruction of the Euclidean geometry[49]. To tune the only free parameter of the model, $d^t$, which is the maximal allowed distance between adjacent DNA fragments, we test FLAMINGO on experimental Hi-C data using different values of $d^t$ to select the distance yielding the smallest objective function as the default value (Supplementary Fig. 21b), which is found to be robust across different chromosomes and cell types (Supplementary Fig. 21c). This model demonstrates fast convergence when applied on both simulated data and experimental Hi-C data (Supplementary Fig. 5 and Supplementary Fig. 21a).

FLAMINGO has an intrinsic computational complexity $O(kN^2)$, where $k$ is a down-sampling rate to define the subset ($\Omega$) of DNA fragment pairs (Supplementary Note 1). Thus, FLAMINGO has sufficiently high scalability to predict high-resolution structures for large genomes, where $N$ is large. Moreover, by using the low-rank property of a 3D distance matrix, FLAMINGO can reconstruct 3D genome structures using a small down-sampling rate $k$, such as 0.2, which can substantially accelerate the optimization. Furthermore, the parallelized computation enabled by the hierarchical prediction strategy further boosts the reconstruction speed.

**Assemble predicted structures from different scales.** The same low-rank matrix completion algorithm is applied separately at two scales: (1) the 1 Mb domain-level scale; and (2) the 5 kb intra-domain scale. To construct the final 3D structure, the predicted intra-domain structures are assembled into the skeleton specified by the domain-level structures. At each 1 Mb domain-level DNA fragment, the center of the corresponding intra-domain structure is assigned at the 3D coordinates predicted for the domain-level fragment. The assigned intra-domain structures are then rotated to minimize the overall reconstruction error between the predicted and the observed pairwise distances over DNA fragments across adjacent domains (inter-domain fragment distances) (Supplementary Fig. 3). To identify the optimal 3D rotation matrices and control the corresponding computational cost, we search for a series of optimal 3D Givens rotation matrices on each dimension. The 3D rotation matrices are then approximated by the multiplication of the 3D Givens rotation matrices.

Denote the predicted intra-domain structure for domain $i$ as $\mathbf{S}_i$. The optimal 3D Givens rotation matrices for the $x$-axis across domains are identified by:

$$\min_{\theta_x^i} \sum_{j,k} \left( ||\mathbf{r}_{\theta_x^i}(\mathbf{S}_{i,j} - \mathbf{C}_i) + \mathbf{C}_i - \mathbf{S}_{i+1,k}||^2 - D_{i,j;i+1,k} \right)^2$$
(9)

where $\mathbf{r}_{\theta_x^i}$ is the 3D Givens rotation matrix of $\mathbf{S}_i$ for the $x$-axis with parameter $\theta_x^i$, $\mathbf{S}_{i,j}$ represents the DNA fragment $j$ within domain $i$, $\mathbf{C}_i$ represents the center of domain $i$ (which is inferred from the domain-level prediction), and $D_{i,j;i+1,k}$ represents the observed squared 3D distance between two inter-domain DNA fragments (fragment $j$ of domain $i$ and fragment $k$ of domain $i+1$) from adjacent domains. The same algorithm is applied to all domains consecutively to search for the rotation matrices of the $x$-axis for all domains. Intuitively, the objective function searches for the best rotation $\mathbf{r}_{\theta_x^i}$ of domain $i$ around its center $\mathbf{C}_i$ to match the distances between fragments across adjacent domains observed from the Hi-C data. The rotation matrices for the $y$-axis and $z$-axis are obtained similarly. Therefore, a series of 3D Givens rotation matrices are identified iteratively for the three axes. Multiplying the converged 3D Givens rotation matrices together yields the optimal 3D rotation matrices which are used to rotate the intra-domain structures, leading to the final genome structure. Since it jointly models all inter-domain distances between adjacent domains (i.e. off-diagonal points) and robustly identifies the global optimal rotation matrices for all intra-domain structures, the rotation algorithm will better align reconstructed structures with the Hi-C data and boost the accuracy of reconstruction.

**Benchmark performance using simulated genome structures.** To quantitatively benchmark the accuracy of FLAMINGO, we simulated 3D genome structures and generated matrices of squared pairwise distances between DNA fragments. The FLAMINGO algorithm was then applied to the squared pairwise distance matrices to reconstruct the 3D structures. The model performance was evaluated by comparing the reconstructed structure with the original structure in two ways. (1) The relative error between the reconstructed 3D coordinates ($\mathbf{C}_{re}$) and the benchmark coordinates $\mathbf{C}_{benchmark}$ of DNA fragments was calculated: $RE_{coord} = ||\mathbf{C}_{re} - \mathbf{C}_{benchmark}||_2^2/||\mathbf{C}_{benchmark}||_2^2$. (2) The relative error between the reconstructed pairwise distance matrix ($\mathbf{R}$) and the original squared distance matrix ($\mathbf{D}$) was calculated: $RE = ||\mathbf{R} - \mathbf{D}^{(1/2)}||_2^2/||\mathbf{D}^{(1/2)}||_2^2$. Moreover, Spearman correlations between predicted and benchmark structures were also calculated to quantify the accuracy.

To test the performance of FLAMINGO with respect to missing data, we randomly down-sampled subsets of the squared pairwise distances as inputs and considered other squared pairwise distances as missing. Multiple down-sampled datasets were generated with different fractions of missing data in terms of different down-sampling rates. FLAMINGO was applied to these down-sampled squared pairwise distance matrices, and the resulting 3D coordinates of DNA fragments were used to calculate the relative errors and correlations.

To further test the performance of FLAMINGO on noisy inputs, we added two levels of white noise separately into the down-sampled squared pairwise distance matrices. As suggested by previous research[49], the first level of noise (Noise level 1) was generated by the normal distribution $N(\delta, \delta)$, where $\delta$ represents the minimum value from the down-sampled squared pairwise distances. Similarly, the second level of noise (Noise level 2) was generated by the normal distribution $N(2\delta, \delta)$. In this way, the noisy down-sampled squared pairwise distances remain positive with high probability, consistent with the basic property of Euclidean distances. The simulations and down-sampling procedures were repeated 10 times for each benchmark setting.

To test the assembly algorithm, we divided the benchmark structure into different domains or fragments. The intra-domain structures were reconstructed separately and then assembled for the final structures, which were compared with the benchmark structure. The relative errors of pairwise distances and 3D coordinates were calculated to demonstrate the high accuracy of the assembly algorithm and its robustness with respect to different choices of domain partitions (Supplementary Fig. 7).

**Performance comparison based on experimental Hi-C data.** For each of the six cell-types, we reconstructed the 3D structures using FLAMINGO at 5 kb-resolution for each of the 23 chromosomes, based on the normalized Hi-C input datasets. To quantitatively evaluate the global reconstruction accuracy of FLAMINGO, we calculated the Spearman correlation coefficients between reconstructed and observed 3D distances for all pairs of DNA fragments, which are defined as all-points correlations. To further evaluate the accuracy of reconstructed intra-domain structures, we also calculated intra-domain correlations based on pairs of DNA fragments within the same domains. An accurately reconstructed structure is expected to demonstrate high correlations, at both all-point and intra-domain levels, which further suggest that the reconstructed structure quantitatively aligns with the observed Hi-C datasets.

We compared the performance of FLAMINGO with seven representative state-of-the-art algorithms:ShRec3D[33], GEM-FISH[34], Hierarchical3DGenome[35], SuperRec[37], ShNeigh[38] and RPR[36]. These methods were selected because they have been shown in previous studies to perform better than other methods using similar modeling strategies, and other existing methods are not included in the comparison because either they have been shown to have less accurate performance by previous studies or they do not practically converge at 5 kb-resolution in our tests. All these methods were applied, based on their suggested parameters, on all of the 23 chromosomes in the six cell-types at 5 kb resolution (Supplementary Note 1). GEM-FISH only finished for chromosome 21. ShRec3D, ShNeigh and RPR finished predictions only for short chromosomes (ShRec3D: chr13–22, ShNeigh: chr15–22 and chrX, and RPR: chr17–22). Hierarchical3DGenome and SuperRec finished predictions for all 23 chromosomes. The correlation coefficients based on those chromosomes with complete predictions were calculated using the same method as explained above. At 5 kb-resolution, the run-times on an AMD EPYC processor with 25 cores were recorded. The maximum memory was set to be 100GB, sufficient for all algorithms.

To further quantify the performance of FLAMINGO with respect to large fractions of missing data, we randomly down-sampled the squared pairwise distance matrix with different down-sampling rates. Using the down-sampled input data, we tested the performance of FLAMINGO and other methods based on the correlation metrics described above. For each down-sampling rate, ten random samples with missing data were generated. The correlation coefficients were calculated for each random sample to evaluate the model performance. Because of impractically long computational times needed by other methods for large chromosomes, only the chromosomes with complete predictions from these methods are included in this comparison.

As orthogonal biological information for model comparisons, we also collected significant long-range chromatin interactions profiled from different experiments, including ChIA-PET[72], Capture-C[45], and SPRITE[46]. For each chromatin interaction, we calculated the predicted 3D distances between the interacting DNA fragments from different reconstruction algorithms. Since interacting DNA fragments (anchors) are close to each other in 3D space, the algorithm is considered to have higher accuracy if it yields shorter predicted distances between interacting DNA fragments.

**Analysis of multi-way chromatin interactions and QTLs.** The multi-way chromatin interactions in GM12878 are collected from a dataset of SPRITE experiments[46]. To identify significant multi-way interactions, Market-Basket algorithm is used to search for higher-order associations of multiple genomic regions that are supported by SPRITE sequencing reads. Significant 3-way, 4-way and 5-way interactions are called based on confidence threshold = 0.1 and support thresholds = $3 \times 10^{-4}$, $2 \times 10^{-4}$ and $1.7 \times 10^{-4}$, respectively. The support thresholds are selected based on the curves of called significant multi-way interactions as a function of different thresholds, and the values corresponding to the elbow points are chosen. Genomic-distance controlled random samples of multi-way interactions are used to generate the background null distribution for statistical testing on the spatial distances among multi-way interacting anchors from the SPRITE data. To compare the fractions of SPRITE multi-way interactions captured by short predicted distances from FLAMINGO versus the fractions captured by short distances converted from Hi-C contact maps, distances are normalized by $F$-norm to guarantee fair comparisons. A variety of thresholds of distances are used to define 3D spatial neighborhoods. A multi-way interaction is considered to be captured if all interacting anchors are located in the same 3D spatial neighborhood. The eQTL datasets[58–61] and hQTL datasets[62] are collected from matched cell-types, including whole blood cells and lymphoblastoid cells. The same normalization procedure is applied to compare the capability of assigning short spatial distances for QTLs based on the predicted distances versus the distances converted from Hi-C contact maps. Similarly, a variety of thresholds of distances are used to define 3D spatial neighborhoods. And long-range eQTLs (>900 kb) and distal hQTLs are evaluated

whether it can be interpreted using the predicted spatial proximity by checking whether the SNP and the target region (i.e. a gene's promoter or histone modification peak) are predicted with shorter spatial distances, compared to samples of genomic-distance controlled random pairs. For every QTL, 1000 random genomic-distance controlled pairs from the same chromosome are generated for comparison.

**Curvature analysis for predicted 3D genome structures.** To calculate the curvature in each 5 kb genomic bin, a quadratic parametric function was fitted locally based on the specific genomic bin and the two neighboring upstream/downstream bins. Assume the parametric representation of the curve is $\vec{r}(t) = (x(t), y(t), z(t))$, where each dimension can be written as a quadratic function, e.g. $x(t) = a_0 + a_1 t + a_2 t^2$. By fitting the curve locally, the curvature is calculated as $\kappa = |\vec{r}'' \times \vec{r}'| / |\vec{r}'|^3$. To have a fair comparison across different chromosomes, curvatures are normalized by the median values of each chromosome. Curvature is then calculated around TAD boundaries[10].

**Comparison with image-based single-cell structures.** 3D coordinates of genomic bins at 30 kb-resolution across single cells for a 2 Mb region in chromosome 21 are collected[68] and compared with FLAMINGO's predictions. In K562, 797 single cells are kept for comparison by filtering out cells with >10% bins having no data (missing data). Linear interpolation is used to fill the missing coordinates in each single cell. To normalize the scales of structures, the 3D coordinate matrix ($\mathbf{P}$) of every single cell (30 kb-resolution) is centered, and then scaled by the $F$-norm: $\mathbf{P}_{scaled} = \mathbf{P}/||\mathbf{P}||_F$. Singular value decomposition (SVD) is then used to rotate and align the normalized single-cell structures (Supplementary Note 1). The average structure across single cells is calculated by taking the mean coordinates for each genomic bin. The predicted consensus structure by FLAMINGO (5 kb-resolution) is centered, scaled and rotated using the same procedure, and is then aligned with the average structure of single cells or cluster-specific average structures. A 30 kb-resolution version of the consensus structure is calculated by taking the average coordinates of six consecutive 5 kb-resolution bins. Hierarchical clustering is applied on single-cell structures based on Euclidean distance to classify the ensemble of single cells into clusters, which can systematically represent the structural variabilities across single cells. After aligning the predicted consensus structure with variable single-cell structures, the differences of coordinates along the genomic region are calculated and compared to the intrinsic standard deviations among single cells.

**Cross cell-type prediction of 3D genome structures.** To predict 3D genome structures in cell-types without Hi-C datasets which are defined as target cell-types, we further expand the FLAMINGO algorithm to combine the Hi-C dataset from a source cell-type and the DNase-seq dataset from the target cell-type, resulting in an integrative variant of FLAMINGO, named as iFLAMINGO. Intuitively, the Hi-C data from the source cell-type facilitate the inference of an approximate structure, which is fine-tuned by the cell-type-specific DNase-seq data from the target cell-type.

Based on the observation that 3D distances between interacting DNA fragments are associated with chromatin accessibilities (Supplementary Fig. 19a), we impute the 3D distances between any two DNA fragments in the target cell-type ($D_{i,j}$) based on DNase-seq signals and 1D genomic distances (Supplementary Fig. 19b). The imputation is achieved by fitting a linear regression model in the source cell-type: $D_{i,j} = \alpha_1 S_i + \alpha_2 S_j + \alpha_3 G_{i,j}$, where $\alpha_1$, $\alpha_2$, and $\alpha_3$ are fitting parameters to be determined, $D_{i,j}$ represents the observed distance, $S_i$ represents the DNase-seq signal of DNA fragment $i$, and $G_{i,j}$ represents the 1D genomic distance between DNA fragments $i$ and $j$. Based on the fitted regression model, 3D distances between DNA fragments can be imputed in the target cell-type, using the target cell-type-specific DNase-seq data, which are then summarized into a matrix $\mathbf{E}$. Therefore, the imputed 3D distance matrix $\mathbf{E}$ represents the target cell-type-specific information which can be used to improve the reconstruction of the corresponding 3D structure.

The imputed 3D distance matrix is integrated into the original objective function as a penalization term, so that we will solve the following problem to reconstruct the 3D structure:

$$\min_{\mathbf{P}} \mathrm{Trace}(\mathbf{PP}^T) + \lambda/2||B(\mathbf{PP}^T - d^t \mathbf{I})||_2^2 + \gamma||A(\mathbf{PP}^T) - A(\mathbf{E}^M)||_2^2, \text{subject to } A(\mathbf{PP}^T) = \mathbf{b} \quad (10)$$

where $\gamma$ is the penalization parameter and $\mathbf{E}^M$ is the Gram matrix of the imputed 3D distance matrix ($\mathbf{E}$) for the target cell-type. The penalization term tunes the reconstructed 3D structure in the target cell-type to align with the imputed 3D distances from DNase-seq. Hence, by borrowing information from the source cell-type Hi-C data, iFLAMINGO predicts the cell-type specific 3D genome structures in the target cell-type.

To validate the performance of cross cell-type predictions, iFLAMINGO was applied to 30 source-target cell-type pairs, based on the six cell-types with Hi-C data available. For each source-target cell-type pair, we predicted the 3D genome structure for the target cell-type based on the Hi-C data from the source cell-type and the DNase-seq data from the target cell-type. The reconstructed 3D structures for target cell-types were evaluated by calculating the correlation coefficients between the reconstructed 3D distance matrix and the observed one based on the Hi-C dataset from the target cell-type. As comparisons, we also evaluated the performance using the reconstructed 3D distance matrices solely based on Hi-C

data from the source cell-type, without incorporating the DNase-seq information from the target cell-type.

**Improve the resolution of 3D genome structures**. iFLAMINGO integrates the high-resolution chromatin accessibility data to improve the resolution of predicted 3D genome structures, such as 5 kb-resolution, based on relatively low-resolution Hi-C contact maps, such as 10 kb-resolution. Given a Hi-C contact map at 10 kb-resolution, we divide each 10 kb genomic fragment into two consecutive 5 kb fragments. The 5 kb fragments inherit the same pairwise 3D distances from the original 10 kb fragment. In this way, the $m$ by $m$ 3D distance matrix at 10 kb-resolution is expanded into a $2m$ by $2m$ 3D distance matrix at 5 kb-resolution, which serves as the initial structure for high-resolution reconstruction. The high-resolution DNase-seq datasets of chromatin accessibility are then incorporated to impute the 3D distances between 5 kb DNA fragments, following the same method described above (Supplementary Fig. 19b). By applying the iFLAMINGO algorithm on the expanded 3D distance matrix from a low-resolution Hi-C contact map and the imputed one from a high-resolution DNase-seq dataset, the 3D genome structure at 5 kb-resolution is then reconstructed. We applied the model on the Hi-C dataset in GM12878 for all of 23 chromosomes at resolution of 10 kb, 25 kb, and 50 kb, respectively. The model performance is evaluated using the correlation coefficients (all-points and intra-domain) between the reconstructed and the observed 3D distance matrices at 5 kb-resolution.

**Statistics and reproducibility**. In performance comparison, all methods are independently applied to reconstruct the 3D structures of the 23 chromosomes in the human genome. The sample sizes of different algorithms are determined by the number of completed reconstructions. In the analyses of ChIA-PET, Capture-C, SPRITE, eQTLs, hQTLs, and single-cell chromatin tracing data, the sample sizes are determined by the original datasets and no data points are excluded from the analysis. No statistical method was used to predetermine sample size. In the simulation analysis and the down-sampling analysis, the performance is evaluated based on ten randomly down-sampled datasets and no data points are excluded from the analyses. To reproduce the analysis, 3D structures of 23 chromosomes in six cell-types and the simulated datasets are provided (see Data availability and Code availability).

**Reporting summary**. Further information on research design is available in the Nature Research Reporting Summary linked to this article.

## Data availability

Hi-C data, along with the annotations of chromatin compartments and TADs, used in this study are available under GSE63525. Capture-C data are available under GSE86189. ChIA-PET data in K562 are available under GSE33664 and ChIA-PET data in GM12878 are available under GSE127053. SPRITE data are available under GSE114242. DNase-seq data are available from ENCODE and Roadmap consortia (https://egg2.wustl.edu/roadmap/web_portal/processed_data.html). ChIP-seq datasets are available from the GEO database (CTCF: GSM822312 [https://www.ncbi.nlm.nih.gov/geo/query/acc.cgi?acc=GSM1002651]; SMC3: GSM935376; Rad21: GSM935332). The gene annotation is available at GENCODE (https://www.gencodegenes.org/human/release_17.html). The eQTL datasets are available from the supplementary data of Battle et al.[58], the MuTHER consortia (http://www.muther.ac.uk/Data.html), the Geuvadis project (https://www.ebi.ac.uk/arrayexpress/experiments/E-GEUV-3/) and the GTEx consortium (https://storage.googleapis.com/gtex_analysis_v7/single_tissue_eqtl_data/GTEx_Analysis_v7_eQTL.tar.gz). The hQTL dataset is available at https://www.zaugg.embl.de/data-and-tools/distal-chromatin-qtls/. Single-cell chromatin tracing data is downloaded from https://github.com/BogdanBintu/ChromatinImaging. The data of simulations, reconstructed chromosome structures, and cross cell-type predictions generated in this study are available in FLAMINGO GitHub repository. Source data are provided with this paper.

## Code availability

The FLAMINGO software[87], along with all the predicted 5 kb-resolution structures of chromosomes 1–22 and X across six cell-types, is available at GitHub. The repository also provides scripts and instructions for structure visualizations, the 1 kb-resolution chromosomal structure predictions for GM12878, sample codes, sample input data, and information of relevant datasets to reproduce the analyses.

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

## Acknowledgements

This work was supported, in part, by awards R01GM131398 and R01AI155775 from the National Institutes of Health and awards NSF2012046 and NSF1942143 from the National Science Foundation. We thank MSU iCER for providing the high-performance computing infrastructure.

## Author contributions

J.Q. and J.W. designed the study and supervised the overall research. H.W. developed the model, implemented the algorithm, and performed data analyses. J.Y. performed data analyses. Y.Z. participated data analyses. H.W., J.Q., and J.W. wrote the manuscript.

## Competing interests

The authors declare no competing interests.
