## [Peer Review File · Nature Communications]

Reviewers' Comments:

Reviewer #1:

Remarks to the Author:

The manuscript by Wang et al. reports a new computational approach to reconstruct the 3D structure of chromosomes using Hi-C data as input. Resolving 3D structure from chromosome conformation capture is a significant problem that is far from resolved and suffers from a lack of algorithmic diversity (as reviewed in doi:10.1093/bfpg/ela004). The authors introduce an innovative approach that borrows from work in other fields, such as recommender systems, where sparse high dimensional data (in this case, the pairwise contacts between each 5k stretch of the chromosome) is assumed to be derived from an underlying low-rank matrix. In this case, the assumption is very well motivated since the high-dimensional Hi-C data does reflect only three dimensions of organization in space.

The authors comprehensively evaluate their method on simulated and experimental data, benchmark it against competing methods, demonstrate its robustness to highly sparse datasets by downsampling, and show its utility in identifying multi-way chromatin interactions. The authors further extend their method to incorporate information about DNA accessibility, allowing it to perform more robustly with sparse data and even impute structure to cell types in which only DNA accessibility is available.

This well-written manuscript provides an impressive and original advance into an important goal. The advantages of their approach, including improved performance and reduced runtime and memory requirements, will make it highly useful for the field. The ability to use sparse Hi-C datasets and incorporate DNase-seq opens the door to many projects since DNase-seq is very widely used and more easily done. Lastly, their approach, which cleverly borrows from an established technique in data science that isn't in widespread use in genomics, will likely inspire future methods in the field.

Overall, I strongly support the publication of this paper following the correction of one major issue outlined below.

Major Comments:

1) The eQTL/hQTL analysis. The method is presented as a way to discover eQTL or hQTL, which is wrong for several reasons (taking eQTL as an example, the below broadly applies to hQTL):

a) eQTLs are associations between a genetic variation and gene expression. They are not binary – the associated allele is important, as is the effect size. None of this could ever be inferred by 3d structure.

b) The way the authors carry the analysis is highly biased and uninformative. Most eQTLs, including in the datasets they use, are found in cis, and most genes have at least one cis eQTL. Therefore, any method that puts a promoter next to its transcript should “discover” eQTLs very well. Of course, this discovery will be uninformative.

While framing the work in the section as QTL discovery is highly misguided and must be removed, their method could still be of great use for interpreting eQTL data. In particular, it could be beneficial in interpreting QTLs found in trans (but on the same chromosome) and even suggest mechanisms, which are largely not known for most eQTLs.

Therefore, I'd like to suggest one of two options:

a) The authors remove the QTL analyses. If the authors do not wish to do more work, I feel the paper stands strong without this section, given everything else they accomplished.

b) The authors limit the QTLs they study to distant QTLs (for example, > 1 MB away from the target gene) and determine how many of the QTLs are inferred to be proximal in 3d location. Such an analysis would be informative for interpreting QTL data and would be attractive to researchers in the eQTL field.

2) Since the authors publish this work as a method, they should organize their code into a usable R package. The code is a series of R scripts with few options to change parameters. Repackaging the code as a documented R package would make a large difference towards making it more usable for the field.

Minor Comment:

1) The authors discuss the algorithm at length in the results sections, with sections lifted almost verbatim from the methods.

2) The authors don't discuss potential future work or limitations. For example, their method be extended beyond chromosome structure to model whole-genome structure? From the methods section, it seems like the main issue could be the regularization parameter that is derived from the "beads on a string" model. I suggest the authors discuss inter-chromosomal structure prediction, either as a limitation of their framework (if it can't be done) or as a potential future extension.

Reviewer #2:

Remarks to the Author:

Title: Reconstruct high-resolution 3D genome structures for diverse cell types using FLAMINGO

Summary

The manuscript proposes a new method FLAMINGO, that provides a scalable reconstruction of 3D genome structures from HiC data in presence of noise and missing data by exploiting low-rank structure present in the pairwise distance matrix between genomic coordinates.

The authors address the following challenges in 3D genome modeling with FLAMINGO along with its integrative version iFLAMINGO: (1) high scalability to reconstruct high-resolution 3D structures genome-wide from massive Hi-C datasets; (2) robust performance to handle large portions of missing data in Hi-C; (3) accurate cross cell-type prediction of 3D structures for cell-types lacking Hi-C datasets; and (4) boosting the resolution of reconstructed 3D structures from low-resolution Hi-C contact maps.

They show this through an extensive evaluation of FLAMINGO and downstream validation of the 3D structures through benchmarking on a simulated dataset, overall 3D structure reconstruction, chromatin loops, TAD, and multiway interactions. Furthermore, the manuscript evaluates the reconstruction performance of FLAMINGO and three competing methods GEM-FISH, ShRec3D, and Hierarchical3DGenome, and shows that FLAMINGO outperforms the competition even with missing different percentages of data from the input HiC data matrix. FLAMINGO uses low-rank matrix completion to offer fast and scalable 3D reconstruction.

iFLAMINGO is a variant of FLAMINGO that integrates low-resolution HiC data or HiC data from a different cell type with 1D epigenomic data like DNase-seq to offer high resolution or cross cell-type 3D reconstruction.

Major:

FLAMINGO makes a claim of state-of-the-art performance on different 3D genome reconstruction challenges. While FLAMINGO is compared to GEM-FISH, ShRec3D, and Hierarchical3DGenome for a subset of experiments performed in the paper, this leaves out a large number of existing methods, such as MCMC5, chromSDE, autochrom3D, Bach, Pastis, InfMod3Gen, MBO, HSA, Chromosome3D, MOGEN, ISDHic, RPR, minimds, SuperRec, MDSGA, GenomeFlow, ShNeigh, and ASHIC. There needs to be a section in the paper describing the innovation offered by FLAMINGO that is absent in these methods. Figures 3c, e, f, and g should be expanded to compare FLAMINGO with these methods and if there are barriers that exist that preclude comparison with certain methods, those reasons should be added to the text.

Figure 6a compares the performance of FLAMINGO across different down-sampling rates and shows that it performs better than Hierarchical3DGenome. However, no reasoning is offered as to why the other two methods (GEM-FISH and ShRec3D) were dropped from this analysis. In figure 6a, the HiC map of Hierarchical3DGenome looks closer to the ground truth (figure 6c) than FLAMINGO at a 5% downsampling rate; this observation is not addressed in the text.

The manuscript claims that more TAD and multiway interactions are captured by the distance matrix generated by FLAMINGO but not the original data, which is to be expected as FLAMINGO is making predictions for missing data in the original HiC matrix. But they don't address the overall quality of the interactions captured by predicted HiC. Some illustrative examples show that they

find interesting interactions but that metrics showing if the model predicts too many false positives or false negatives or other summary statistics are missing. The benchmarking simulated data could be used to assess the quality of discovered interactions.

While the manuscript provides extensive validation of FLAMINGO, the codebase is lacking. There are two aspects of the codebase I want to comment on reproducibility and reusability. On the reproducibility front, currently, it is not possible to replicate the figures shown in the paper using the codebase. Both the data and code that went into making the figures is missing. While the data can be downloaded and download instructions are provided in the repo, the intermediate data files, and code that went into making the figures should be easily accessible for further inspection. On the reusability front, the codebase is extremely poor. The main value add of a new method is that people should be able to use it on new datasets, but the current code structure is too spaghetti-like and would probably break if tried on new data. Furthermore, the HiC data is stored in a custom sparse format which further hinders reusability. There are several standard HiC data formats like .HiC, .cool which would increase ease of use of the method.

Minor:

The mathematical equations in the methods section are embedded in the paragraphs making them hard to read.

Reviewer #3:

Remarks to the Author:

The authors present a computational method named FLAMINGO in enabling high-resolution reconstruction of single chromosomes from Hi-C across multiple cell types at a high computing efficiency. The method was built base on low-rank matrix algorithms and benchmarked against several published models on different metrics on chromatin structural reconstruction. The authors did a tremendous amount of work from different aspects, including structural reconstruction, benchmarking, application over multiple cell lines, scalability studies, higher-order interaction and QTLs discoveries, single-cell interpretations, and Hi-C resolution improvement. However, as detailed below, there are quite a number of technical barriers, overstating problems, and inadequate writing that prevent me from recommending the publication of this paper in Nature Communications.

Major issues:

1. The low-rank matrix algorithm is at the core of the model. The authors need to elaborate more on the theoretical details of a number of statements in the manuscript. For example, it would be helpful to give at least a sketch on the mathematical intuition behind "...the resulting symmetric Euclidean distance matrix has a rank at most 5", but not simply have a reference there. If they want to have a broad audience, this is not quite a responsible way of doing things.
2. Along the manuscript, the Hi-C map and distance map seemed to be used interchangeably. Although the authors provided evidence to show that the threshold of converting from distance map to contact map is insensitive, they needs to be very clear on the concept that these two things are far away from each other [PMID: 25512564, PMID: 26223848, PMID: 27760553, PMID: 28604723]. One need to be paying more attention especially when the resolution of the model is higher. I doubt the current model could capture any of these important observations, as the root between the inconsistency of the Hi-C contacts and spatial distance might come from the structural heterogeneity. But if the authors have anything else to show, it would be necessary to show to make this important distinction clear.
3. How do the authors treat the centromeric regions, where a large chunk of Hi-C data is missing, in terms of the modeling and algorithmic aspect?
4. In figure 1b (and figure S10), the authors showed a representative CTCF-mediate loop. However, this is far from enough. A more systematic characterization of how well/bad the model is able to recapitulate the CTCF-loop is needed.

5. On page 7 in the section "Benchmark performance based on simulated structures", the authors were comparing the performance of the model with benchmarked simulated structures. This is a useful comparison, but I wonder would the performance be affected by the length of the chain? For example, regarding the longest chromosome - chr1 - which has 44k fragments in the current setting (why 44k but not 50k? a separate question below), would you still get such a good benchmark performance? It seems that the benchmark system in figure 2 is rather short, but it is important to benchmark on a higher level.

6. On page 8 in the section "Superior reconstruction accuracy across diverse cell-types", The authors claimed a high correlation (0.95) for the skeleton prediction. This number, though looks high, probably does not say anything. With the current resolution, it is easy to achieve this kind of number. If the authors could show a good agreement between the prediction and Hi-C in a sequence-dependent manner, especially at the very off-diagonal region, it is more convincing and informative than the numbers shown here.

7. At the same place, the authors claimed the numbers are "consistently higher than other methods". I assume these "other methods" are a few ones later on mentioned (e.g. Hierarchical3DGenome and such), but I am seeing it correctly, they were not introduced at all till such point. It led to a lot of confusion when I was reading this.

8. Again, in the same paragraph, the authors claimed that "FLAMINGO reconstructs clear loop structures for TADs and predicts short 3D distances for inter-TAD chromatin contacts". However, from the Hi-C, it doesn't seem that the Hi-C shows a visibly strong signal of an off-diagonal contact indicating the loop, which was seen strongly in the predicted map (figure 3a map). Could the authors comment on this inconsistency?

9. On page 9, the authors compared the Spearman correlations between predicted and observed 3D distances and generated a number. Again, this number is less useful compared to a sequence-dependent correlation, as a good correlation score might mainly coming from the diagonal values, but I guess most of the available models are good in near-diagonal prediction but the off-diagonal values are more important and more difficult to capture.

10. On page 10, the authors did cross-comparison between different methods in capturing loci proximity from experimental values and showed their method consistently predicts a small pair-wise distance than other ones. However, I didn't see any direct comparison between the authors' model and experimental values.

11. On the same page, the authors compared one enhancer-promoter pair. This itself is a non-trivial question. If the authors want to make a good claim on this, they need to do a more comprehensive characterization of E-P pairs (there are public datasets on this, e.g. PMID: 27064255) globally instead of only showing one or two pairs and make such an important statement. For example, what about E-P pairs without CTCF anchors? Is there any difference between these with the ones with CTCF? And if their model is really powerful in capturing those, of all the E-P pairs, what is the false positive/false negative rate of prediction?

12. On the same page in section "Advanced scalability for large-scale chromosome conformations", given the authors' statement, I found it would interesting to show the model's capability at 1kb if the authors really want to demonstrate the power of the model as they claimed, given that 1kb is the highest resolution with the bulk Hi-C data available, and also as probably known by the authors, 5kb, though indeed very high, is in principle not too impressive as there are other studies already achieving structural predictions at similar levels. I acknowledge that 1kb might be the hallmark at this point given all experiments we have and can be computationally challenging based on what the authors claimed, but if it can be shown on a smaller genomic region, e.g. 40Mb (~40k fragments), which is still similar to what the authors are doing here in terms of the computational complexity, that would be more interesting and also kind of a requirement to fulfill the authors' claim on the superior of the model in dealing with data sparsity/noise and high efficiency.

13. Also, given that the current pipeline runs for only 42 mins for 5kb on chromosome 1 and the

computational complexity is $O(N^2)$, if 1kb is used that would presumably be 25 times slower, which is still acceptable on things like HPCC if available. Anyway, I don't see any reason of not going into a higher resolution if the advantage of the model that the authors claimed are really significant to be marked.

14. Similar to comment 10, I didn't see any direct comparison between model results and experiments (e.g. SPRITE) in multi-way interactions, besides cross-comparison of different computational methods. Also, in figure 4a, first panel, is it really significant (the authors marked *** on the top)? From what I see, the two bars are almost aligned to each other.

15. Similar to comment 2, on page 15, the authors wrote that "it suggests that the distance information derived from Hi-C contact frequency is consistent with the spatial configurations obtained from imaging techniques". This wording itself is problematic. The authors need to be careful.

16. In the section "Reference structure to interpret single-cell variabilities" on page 15, the authors compared the model results with clusters of single-cell structures and found a high similarity of the predicted consensus structure to the averaged cluster 2 structure. Do these results indicate that the consensus structure might only be the representation of one of the local minimums in the entire structural ensemble characterized by the energy landscape, as the chromatin folding has been shown to be very frustrated and glass-like [PMID: 31467267, PMID: 30089831]. This might be reflected from the perspective of optimization in the authors' method, or might be the heterogenous nature of the chromatin population that the consensus structure method is not able to capture. This is an important point and the authors need to be more deliberate.

17. In the section "Cross cell-type prediction of 3D structures", a more quantitative way to reveal the subtle difference would be to look at the correlation between the prediction map- for example, is the correlation between the distance map of K562 (predicted) and K562 (experiment) is higher than the correlation between the distance map of GM12878 (experiment) and K562 (experiment), to show that the model is able to capture the difference between different experiments.

18. I wonder if it is ever possible to apply the model to inter-chromosomal contacts, as they are also available as Hi-C, but of course, worse in terms of the quality and appear to be noisier. I am not an expert in this subfield but there are multiple studies out there that enable the whole-genome structural reconstruction from different perspectives and different types of method [PMID: 26951677, PMID: 25961318, PMID: 33086041]. I feel the authors' model could have potentials on this perspective to generate some impacts. And also, this would not only be a good extension of the model but also a reasonable validation against the authors' statement on the superiority of the model.

19. Lastly, I don't think the authors gave a good discussion. As pointed out by a handful of recent reviews on different methods in chromatin modeling [PMID: 31470090, PMID: 32320675, PMID: 34241389, PMID: 33253996], clear drawbacks exist for the consensus structure method in reconstructing the chromatin structures. For example, it is not possible to study the dynamics, and some basic experimental observations on dynamical looping and neighboring TAD interactions cannot be revealed. However, the authors did not provide any substantial discussion on these aspects. The impression I got after reading is that they claim the model is totally the "supreme" one than anything else, but as detailed above, that's probably not true.

Minor issues:

1. Some portions of the abstract need to be rewritten. For example, key algorithm and key model innovation should be highlighted. Though it was mentioned that "compressing inter-dependent Hi-C interactions to delineate the underlying low-rank structures in 3D space", the statement remains allusive in revealing the core algorithm of the implementation.

2. The authors need to be more careful in their word choice. For example, please avoid using things like "whole human genome", as it is very easy to give people the impression that it is the

real whole-genome involving inter-chromosomal interactions as well. As the authors might know, there is quite an extensive amount of studies over there as well. However, the model is far from a complete "human genome".

3. On page 3, the sentence "The accuracy of a predicted structure is mainly evaluated by its capability of recapitulating the pairwise distances between genomic loci observed from Hi-C, such as correlations". This is a bad sentence, I guess the "correlation" is simply the "evaluation", but it is very confusing as the Hi-C contacts or cross-loci distance is also a "correlation". The last three words are basically meaningless to be here. Also, this sentence (and other places in the manuscript) is scientifically incorrect: one cannot observe pairwise distance from Hi-C.

4. Why chromosome 1 (~250Mb) with 5kb only has 44k fragments?

Reviewer #4:

Remarks to the Author:

This paper introduces a computational algorithm, FLAMINGO, for 3D reconstruction of the genome from Hi-C contact maps. The novelty of the method over previous algorithms is that the objective function used to predict the 3D coordinates of the genome is low-rank constrained. Thus, solving a low-rank matrix completion formulation, FLAMINGO can perform faster reconstruction from a subset of Hi-C data. The paper first shows that FLAMINGO can accurately model the simulated data. Next, it compares the method to other baselines on real-world datasets across 6 different cell lines. It also demonstrates low runtimes for an increasing number of loci compared to other methods. The subsequent sections use a variety of orthogonal datasets to demonstrate that the reconstructions from FLAMINGO - (1) capture pair-wise and multi-way chromatin interactions (2) capture eQTLs, hQTLs, and TAD curvature (3) are consistent with single-cell structures (4) produce high-quality Hi-C maps. Finally, the paper presents integrative-FLAMINGO (iFLAMINGO), an extension of the work, to incorporate 1D epigenomic signals in the 3D reconstruction to prejudice accurate cross-cell-type predictions.

Overall, with its comprehensive set of results, the paper makes a compelling case for the proposed method. FLAMINGO is faster and more accurate than the existing methods and can assist in a variety of downstream biological applications. However, some points/descriptions in the paper require clarifications (see below) for a better understanding of the work.

Comments:

The choice of the baseline methods and the evaluation metric is not very clear. While the paper includes some recent methods, others like SuperRec (Zhang et al.) were not included. The paper need not include all the methods for comparison but a description of the relevant state-of-the-art would be helpful. Also, some of the previous works have included metrics like RMSD in addition to Spearman correlation for evaluation. Is there a reason why those were not included in the current study? Is correlation sufficient to compare distance matrices that tend to have underlying structural properties?

For a reader not familiar with the 3D reconstruction literature, it is not clear how some of the components of the algorithm compare or connect with the previous literature. How is this work placed with respect to previous works? For example, the hierarchical setup of constructing domain-wise skeleton and filling in intra-domain structures - is it similar to (or different from) the one proposed in Hierarchical3DGenome paper (Trieu et al.)?

Since the reconstructed structures from FLAMINGO assign shorter 3D distances between anchors of chromatin interactions, it would be interesting to see if there are examples (like the one presented for KCNA2) where the baseline methods (like Hierarchical3DGenome) did not capture the close proximity of enhancer and promoter. This result would solidify the importance of FLAMINGO for the downstream investigation of promoter-enhancer interactions.

It is unclear why the Hierarchical3DGenome was not included as a baseline in the results

presented for multi-way interactions, eQTLs, and reconstructions from single-cell (Figures 4 and 5)

Supplementary Figure 17 does not show the values of the objective function for different values of dt parameter, instead, it seems that the value of 0.05 was chosen based on the intra-domain correlation metric. Is this value for dt consistent across all the experiments and all the cell lines?

I could not find any descriptions about the parameters that need to be tuned for the baseline 3D reconstruction methods and whether adequate tuning was performed for them before evaluating their performance with FLAMINGO.

Minor points:

Figure 3(c) compares performances for multiple chromosomes but Supplementary figure 9 presents results for only chromosome 21. Is this because of runtime constraints?

α notation in the Methods is used for two different entities - (1) free parameter for distance matrix calculation (2) index of DNA fragment pair. Please use a different notation for one of them.

The method section description is dense making it hard to follow. It would be useful to separate the equations from the text and provide the equation numbers.

Responses to Reviewer Comments

We really appreciate reviewers' comments and suggestions to improve our manuscript. Here is a **brief highlight of the major revisions and improvements**:

1. We have now revised the writings for the methods section to:

- a) include more explanations on the low-rank property of the distance matrices;
- b) add more clarifications on some concepts, terms and analysis procedures;
- c) re-wrote and shortened the paragraphs on the overview of model frameworks;
- d) all major equations are listed and numbered in separate lines in the methods section to improve the clarity for the readers.

2. We have substantially re-organized the code of FLAMINGO and have created the R package (FLAMINGOr) so that it is easier for the users to install, test and apply FLAMINGO on their own data. The R package supports input data in community-accepted standard formats, such as *.hic*, *.mcool* and the sparse matrix format. The new R package, along with the detailed instructions and some sample inputs, are all posted in the FLAMINGO GitHub repository (<https://github.com/wangjr03/FLAMINGO>).

3. We have now benchmarked FLAMINGO with three more methods (ShNeigh, SuperRec and RPR). Together with the previously included methods (ShRec3D, GEM-FISH and Hierarchical3DGenome), we have compared the performance of FLAMINGO with 6 methods in total, and FLAMINGO still shows the best performance.

We also looked at other methods mentioned by the reviewers, which fall into the following categories: a) have been shown by previous studies to have less accurate performance than the 6 methods that we have included; b) the code is available but does not converge for 5kb-resolution predictions; c) the code is not provided by the original authors; and d) the method solves different problems, such as signal deconvolution, structure visualization or scHi-C analysis. Detailed explanations are provided in the point-by-point responses. So they are not included in the comparisons.

Our expanded method performance comparison analyses are reflected in the revised and new figures:

a) Figure 3c and Supplementary Figure 11 & 12 (accuracy comparison benchmarked by Hi-C data);

b) Figure 3e-g and Supplementary Figure 13 (accuracy comparison benchmarked by orthogonal chromatin interaction datasets of ChIA-PET, Capture-C and SPRITE);

c) Supplementary Figure 14b (computational scalability comparison);

d) Figure 6e and Supplementary Figure 18b-e (comparison on the capability of handling missing data).

4. We have removed the previous claims that our method can predict/discover QTLs. In addition, based on the reviewer's suggestion, we have included new analyses on long-range QTLs that suggest FLAMINGO can help to interpret long-range QTLs. These new results are discussed in the revised paragraph on QTL analysis, and are shown in Figure 4d-f.

5. We have added one new paragraph in Discussion about the current limitations and potential future developments.

6. We have also generated the first 1kb-resolution (the highest resolution to date) 3D structure predictions for all chromosomes of the human genome in GM12878, and evaluated the performance accuracy and computational scalability for 1kb-resolution reconstructions.

These new results of 1kb-resolution predictions have been discussed in the revised manuscript, shown in the new Supplementary Figure 2, and shared in the GitHub repository.

Below are detailed point-by-point responses to specific comments.

REVIEWER COMMENTS

Reviewer #1 (Expertise: eQTL prediction):

The manuscript by Wang et al. reports a new computational approach to reconstruct the 3D structure of chromosomes using Hi-C data as input. Resolving 3D structure from chromosome conformation capture is a significant problem that is far from resolved and suffers from a lack of algorithmic diversity (as reviewed in doi:10.1093/bfpg/elaa004). The authors introduce an innovative approach that borrows from work in other fields, such as recommender systems, where sparse high dimensional data (in this case, the pairwise contacts between each 5k stretch of the chromosome) is assumed to be derived from an underlying low-rank matrix. In this case, the assumption is very well motivated since the high-dimensional Hi-C data does reflect only three dimensions of organization in space. The authors comprehensively evaluate their method on simulated and experimental data, benchmark it against competing methods, demonstrate its robustness to highly sparse datasets by downsampling, and show its utility in identifying multi-way chromatin interactions. The authors further extend their method to incorporate information about DNA accessibility, allowing it to perform more robustly with sparse data and even impute structure to cell types in which only DNA accessibility is available. This well-written manuscript provides an impressive and original advance into an important goal. The advantages of their approach, including improved performance and reduced runtime and memory requirements, will make it highly useful for the field. The ability to use sparse Hi-C datasets and incorporate DNase-seq opens the door to many

projects since DNase-seq is very widely used and more easily done. Lastly, their approach, which cleverly borrows from an established technique in data science that isn't in widespread use in genomics, will likely inspire future methods in the field. Overall, I strongly support the publication of this paper following the correction of one major issue outlined below.

Response: We really appreciate the reviewer's comments on the significance, novelty, performance advantages and the wide applicability of our work. We are glad that the reviewer found our work to be impressive and original.

Major Comments:

1) The eQTL/hQTL analysis. The method is presented as a way to discover eQTL or hQTL, which is wrong for several reasons (taking eQTL as an example, the below broadly applies to hQTL):

a) eQTLs are associations between a genetic variation and gene expression. They are not binary – the associated allele is important, as is the effect size. None of this could ever be inferred by 3d structure.

b) The way the authors carry the analysis is highly biased and uninformative. Most eQTLs, including in the datasets they use, are found in cis, and most genes have at least one cis eQTL. Therefore, any method that puts a promoter next to its transcript should “discover” eQTLs very well. Of course, this discovery will be uninformative.

While framing the work in the section as QTL discovery is highly misguided and must be removed, their method could still be of great use for interpreting eQTL data. In particular, it could be beneficial in interpreting QTLs found in trans (but on the same chromosome) and even suggest mechanisms, which are largely not known for most eQTLs. Therefore, I'd like to suggest one of two options:

a) The authors remove the QTL analyses. If the authors do not wish to do more work, I feel the paper stands strong without this section, given everything else they accomplished.

b) The authors limit the QTLs they study to distant QTLs (for example, > 1 MB away from the target gene) and determine how many of the QTLs are inferred to be proximal in 3d location. Such an analysis would be informative for interpreting QTL data and would be attractive to researchers in the eQTL field.

Response: Thanks for this important comment. We agree with the reviewer's comment and appreciate the suggestions to re-frame the analysis as interpreting distant QTLs. We have now removed the previous claims about discovering/predicting QTLs.

Based on the suggestion (b), we have now included new analyses on long-range eQTLs and distal hQTLs. We have tried to find datasets that contain eQTLs that are >1Mb away, but none of them provide such cases. Therefore, we restricted our eQTL analysis to the ones that are >900kb (identified in the GTEx project) and the distal

hQTLs identified from *Grubert et al., 2015 Cell*. Although they are not >1Mb away, they can still be considered to be distant QTLs.

For each distant QTL, we generated 1,000 random SNP-target pairs with the same genomic-distance and from the same chromosome as the background. Overall, 671 (54.7%) distal eQTLs and 11,797 (56.3%) distal hQTLs are predicted to have spatial 3D distances that are at least 2-fold shorter than the median distances of their corresponding random samples. In addition, the overall distribution of predicted spatial distances of distant QTLs are significantly shorter than the distribution from the genomic-distance controlled random SNP-target pairs (p -value=1.3x10⁻³ for eQTLs and p -value=2.84x10⁻³ for hQTLs, one-sided Wilcoxon test).

The new statistical tests and individual examples based on distant eQTLs and hQTLs are shown in Figure 4d-g. The previous results are moved to Supplementary Figure 16.

The 2nd paragraph of the section “Analysis of multi-way interactions and QTLs by FLAMINGO beyond 2D Hi-C contact maps” is substantially revised to incorporate these new results and also the analysis of specific examples (e.g. a 983kb eQTL showing short spatial distance by FLAMINGO, Fig. 4e). This paragraph is now re-framed to suggest FLAMINGO’s ability in helping to interpret the potential mechanisms of distal QTLs.

2) Since the authors publish this work as a method, they should organize their code into a usable R package. The code is a series of R scripts with few options to change parameters. Repackaging the code as a documented R package would make a large difference towards making it more usable for the field.

Response: We agree with the reviewer’s comment. We have re-organized the code into a new R package (FLAMINGOr), which can be easily installed, tested and applied on new data. The new R package, along with the instructions and some sample inputs, are posted in the FLAMINGO GitHub repository (<https://github.com/wangjr03/FLAMINGO>). The users can run FLAMINGO using a single command on their data (as shown by the example code in the GitHub repository), and the package can support different community-accepted standard formats of input data.

The previously posted R scripts are also kept in the GitHub repository, in case the users want to test and analyze different components of our method.

Minor Comment:

1) The authors discuss the algorithm at length in the results sections, with sections lifted almost verbatim from the methods.

Response: Thanks for pointing this out. We have now revised and shortened the related paragraphs in the Methods section (under the section “Model framework of FLAMINGO”) to reduce the potential redundancy. The paragraphs in the Results section are largely

maintained because some concepts and the algorithm design need to be introduced to the readers at an early stage in the paper.

2) The authors don't discuss potential future work or limitations. For example, their method be extended beyond chromosome structure to model whole-genome structure? From the methods section, it seems like the main issue could be the regularization parameter that is derived from the "beads on a string" model. I suggest the authors discuss inter-chromosomal structure prediction, either as a limitation of their framework (if it can't be done) or as a potential future extension.

Response: Thanks for the suggestion. We have now added one paragraph in the Discussion section about current limitations and future developments (4th paragraph in the Discussion section). There are currently two major limitations for future developments. First, to predict whole-genome structures, unlike intra-chromosomal structures, a reliable conversion function is needed to convert inter-chromosomal Hi-C interaction frequency to spatial distances, which is currently not available. Second, to predict single-cell level chromosomal structures, the method needs to be further developed to handle the extremely high rates of missing data of single-cell Hi-C datasets. For bulk tissue Hi-C data, FLAMINGO has been shown to robustly handle 98% missing data at 5kb-resolution. In contrast, a typical single-cell Hi-C data has >99.99% missing data even at 100kb-resolution. This is a big challenge for the field to address.

Reviewer #2 (Expertise: genome organization, ML):

Title: Reconstruct high-resolution 3D genome structures for diverse cell types using FLAMINGO

Summary

The manuscript proposes a new method FLAMINGO, that provides a scalable reconstruction of 3D genome structures from HiC data in presence of noise and missing data by exploiting low-rank structure present in the pairwise distance matrix between genomic coordinates.

The authors address the following challenges in 3D genome modeling with FLAMINGO along with its integrative version iFLAMINGO: (1) high scalability to reconstruct high-resolution 3D structures genome-wide from massive Hi-C datasets; (2) robust performance to handle large portions of missing data in Hi-C; (3) accurate cross cell-type prediction of 3D structures for cell-types lacking Hi-C datasets; and (4) boosting the resolution of reconstructed 3D structures from low-resolution Hi-C contact maps.

They show this through an extensive evaluation of FLAMINGO and downstream validation of the 3D structures through benchmarking on a simulated dataset, overall 3D structure reconstruction, chromatin loops, TAD, and multiway interactions. Furthermore,

the manuscript evaluates the reconstruction performance of FLAMINGO and three competing methods GEM-FISH, ShRec3D, and Hierarchical3DGenome, and shows that FLAMINGO outperforms the competition even with missing different percentages of data from the input HiC data matrix. FLAMINGO uses low-rank matrix completion to offer fast and scalable 3D reconstruction.

iFLAMINGO is a variant of FLAMINGO that integrates low-resolution HiC data or HiC data from a different cell type with 1D epigenomic data like DNase-seq to offer high resolution or cross cell-type 3D reconstruction.

Response: We really appreciate the reviewer's positive comments on the novelty and the demonstrated advantages of our algorithm.

Major:

FLAMINGO makes a claim of state-of-the-art performance on different 3D genome reconstruction challenges. While FLAMINGO is compared to GEM-FISH, ShRec3D, and Hierarchical3DGenome for a subset of experiments performed in the paper, this leaves out a large number of existing methods, such as MCMC5, chromSDE, autochrom3D, Bach, Pastis, InfMod3Gen, MBO, HSA, Chromosome3D, MOGEN, ISDHiC, RPR, minimds, SuperRec, MDSGA, GenomeFlow, ShNeigh, and ASHiC. There needs to be a section in the paper describing the innovation offered by FLAMINGO that is absent in these methods. Figures 3c, e, f, and g should be expanded to compare FLAMINGO with these methods and if there are barriers that exist that preclude comparison with certain methods, those reasons should be added to the text.

Response: Thanks for this important comment. We agree with the reviewer that an expanded method comparison analysis is helpful to further support FLAMINGO's superior performance. Some of the algorithms mentioned by the reviewer were not included in the original version of the manuscript because they have been shown by previous studies to have less accuracy compared to Hierarchical3DGenome, GEM-FISH or ShRec3D.

We have looked at all these methods carefully (all these 3D reconstruction methods are now cited in Introduction) and expanded our performance comparisons by further including SuperRec, ShNeigh and RPR. Therefore, we have compared FLAMINGO with 6 methods in total, and FLAMINGO still shows the best performance: 1) higher accuracy benchmarked by measured Hi-C; 2) higher accuracy benchmarked by orthogonal chromatin interaction data, including ChIA-PET, Capture-C and SPRITE; 3) better computational scalability; and 4) better capability to handle high rates of missing data.

These new results of expanded method comparisons are now discussed in different parts of Results:

a) the 3rd and 4th paragraphs in the section "Superior reconstruction accuracy across diverse cell-types";

b) the 1st paragraph in the section "Advanced scalability for large-scale chromosome conformations";

c) the 3rd paragraph in the section “Robust performance to handle missing data in Hi-C datasets”,

The method comparison part in the Methods is also revised accordingly, along with the brief explanation on the selection criteria, see:

the 2nd paragraph in the section “Performance comparison based on experimental Hi-C data”.

These expanded performance comparison results are shown in the revised or new figures:

a) Figure 3c and Supplementary Figure 11 & 12 (accuracy comparison benchmarked by Hi-C data);

b) Figure 3e-g and Supplementary Figure 13 (accuracy comparison benchmarked by orthogonal chromatin interaction datasets of ChIA-PET, Capture-C and SPRITE);

c) Supplementary Figure 14b (computational scalability comparison);

d) Figure 6e and Supplementary Figure 18b-e (comparison on the capability of handling missing data).

The methods that are not included in the comparison analysis fall into the following 4 categories of reasons:

a) The methods have been shown in previous studies that they perform less accurately than the ones that we have included. This category includes: chromSDE, HSA, MOGEN, and miniMDS. These methods have been compared by SuperRec, Hierarchical3DGenome and LordG (LordG is an earlier version of Hierarchical3DGenome). This category also includes GenomeFlow, which is a GUI of LordG.

b) The methods provide code but the code has bugs or can not practically converge at 5kb-resolution. This category includes: MDSGA, autochrom3D, PASTIS, MCMC5 and BACH.

c) The methods do not provide any code to run. This category includes: MBO (the link of the code already expired).

d) The methods require different types of input features or target different scientific problems. This category includes: ASHiC (deconvolution), ISDHi-C (for single-cell Hi-C), InfMod3Gen (needs significant chromatin interactions as inputs, instead of Hi-C contact maps)

The innovation of the model design and algorithmic advantages of FLAMINGO are summarized in the 3rd paragraph of Introduction, the 2nd paragraph of the section “FLAMINGO algorithm to reconstruct high-resolution 3D genome architectures” of Results, and the 1st paragraph of Discussion.

Figure 6a compares the performance of FLAMINGO across different down-sampling rates and shows that it performs better than Hierarchical3DGenome. However, no reasoning is offered as to why the other two methods (GEM-FISH and ShRec3D) were dropped from this analysis.

Response: Thanks for this great suggestion. We have now included all the other methods for this comparison (Hierarchical3DGenome, GEM-FISH, ShRec3D, ShNeigh, SuperRec and RPR), and FLAMINGO demonstrates the best performance in handling high rates of missing data. The results are shown in the revised Figure 6e and Supplementary Figure 18b.

In addition, we have also expanded the comparison using other types of chromatin interactions (ChIA-PET, Capture-C and SPRITE) to benchmark this comparison, when 50% of observed Hi-C signals are down-sampled as inputs. FLAMINGO also shows the best accuracy, which are shown in the new Supplementary Figure 18c-e.

In figure 6a, the HiC map of Hierarchical3DGenome looks closer to the ground truth (figure 6c) than FLAMINGO at a 5% downsampling rate; this observation is not addressed in the text.

Response: As a clarification, at 5% down-sampling rate, the predicted structure by Hierarchical3DGenome is not closer to the ground truth and it contains substantial intra-domain fluctuations without a clear structural configuration (Figure 6a), especially if it is compared to the structure predicted also by Hierarchical3DGenome at 70% (Figure 6a). The TAD structures are largely disrupted at 5% vs. 70% from Hierarchical3DGenome's predictions, but are relatively better maintained from FLAMINGO's predictions.

The resulting Hi-C map (by Hierarchical3DGenome at 5%) contains large portions of almost identical intra-domain contacts, and that is why there are many intra-domain fluctuations in the predicted structure, because almost every locus has similar distances to many other loci within the same domain. In addition, the two TADs on the right corner of the map are almost merged together, while there is a clear partition between these 2 TADs as shown in Figure 6c.

As the quantitative metric, the distance matrix generated by FLAMINGO for this genomic region (Fig. 6a), at 5% down-sampling rate, has a Spearman correlation of 0.643 with the observed matrix (Fig. 6c), while the matrix generated by Hierarchical3DGenome only achieves a Spearman correlation of 0.497.

The manuscript claims that more TAD and multiway interactions are captured by the distance matrix generated by FLAMINGO but not the original data, which is to be expected as FLAMINGO is making predictions for missing data in the original HiC matrix. But they don't address the overall quality of the interactions captured by predicted HiC. Some illustrative examples show that they find interesting interactions

but that metrics showing if the model predicts too many false positives or false negatives or other summary statistics are missing. The benchmarking simulated data could be used to assess the quality of discovered interactions.

Response: Thanks for this comment. As clarification, the overall metrics are actually provided in Figure 4a. In Figure 4a, we have shown that, compared to rigorous genomic-distance controlled random multi-way interactions, the experimentally identified multi-way interactions (by SPRITE) are predicted with significantly shorter spatial distances by FLAMINGO. These significant differential distributions (Fig. 4a) provide the global quality metrics with respect to FLAMINGO's ability of capturing multi-way interactions, especially for 4-way and 5-way interactions, excluding the possibility of too many false positives or false negatives.

While the manuscript provides extensive validation of FLAMINGO, the codebase is lacking. There are two aspects of the codebase I want to comment on reproducibility and reusability. On the reproducibility front, currently, it is not possible to replicate the figures shown in the paper using the codebase. Both the data and code that went into making the figures is missing. While the data can be downloaded and download instructions are provided in the repo, the intermediate data files, and code that went into making the figures should be easily accessible for further inspection. On the reusability front, the codebase is extremely poor. The main value add of a new method is that people should be able to use it on new datasets, but the current code structure is too spaghetti-like and would probably break if tried on new data. Furthermore, the HiC data is stored in a custom sparse format which further hinders reusability. There are several standard HiC data formats like .HiC, .cool, which would increase ease of use of the method.

Response: Thanks for this important comment and we appreciate the suggestions. We have revised the code of FLAMINGO and updated the GitHub repository. Below is a summary of the updates:

1. The code is re-organized and a single R package (FLAMINGOr) is created, so that the users can easily install and test. Users can run FLAMINGO using just one command line on their data (by running 'flamingo.main_func_large()', as explained in the repository's README with examples). The parameters of FLAMINGOr can be easily set.
2. The FLAMINGOr package can support the standard formats of input data, including .hic, .mcool and the sparse matrix formats.
3. The README has been revised accordingly to include the instructions about using FLAMINGOr, along with example code.
4. The figures of predicted 3D structures in the paper are made by the open-source software ParaView. The FLAMINGOr package now provides the function ('write.vtk()') to prepare the format of the predictions that can be readily used by ParaView to visualize (instructions explained in the README at the GitHub repository).
5. In addition to the previously posted 3D structure predictions that can be used to generate most of the figures in the paper, we have also uploaded the benchmark

structures used in the simulation analyses (with different noise and down-sampling rates) and the down-sampled Hi-C matrices as sample inputs onto the GitHub repository, so that users can test FLAMINGO using these sample inputs. The descriptions of these sample inputs are included in the README at GitHub.

6. Other data, such as the original Hi-C, ChIA-PET, Capture-C, SPRITE, eQTLs, hQTLs and STORM datasets, are from other studies and may not be allowed to post at our GitHub repository. We have expanded the instructions on how to find and download those datasets in the README at GitHub.

As a side note, the separate R scripts of FLAMINGO are still kept on the GitHub repository, in addition to the new FLAMINGO R package, because it can be convenient for users to analyze, test and evaluate different parts of the software specifically.

Minor:

The mathematical equations in the methods section are embedded in the paragraphs making them hard to read.

Response: Thanks for the suggestion. We have now listed and numbered the major equations in separate lines.

Reviewer #3 (Expertise: predicting genomic 3D organization):

The authors present a computational method named FLAMINGO in enabling high-resolution reconstruction of single chromosomes from Hi-C across multiple cell types at a high computing efficiency. The method was built base on low-rank matrix algorithms and benchmarked against several published models on different metrics on chromatin structural reconstruction. The authors did a tremendous amount of work from different aspects, including structural reconstruction, benchmarking, application over multiple cell lines, scalability studies, higher-order interaction and QTLs discoveries, single-cell interpretations, and Hi-C resolution improvement. However, as detailed below, there are quite a number of technical barriers, overstating problems, and inadequate writing that prevent me from recommending the publication of this paper in Nature Communications.

Response: We appreciate the reviewer's positive comments on the amount of our work and analyses. We have clarified and addressed the specific comments below.

Major issues:

1. The low-rank matrix algorithm is at the core of the model. The authors need to elaborate more on the theoretical details of a number of statements in the manuscript. For example, it would be helpful to give at least a sketch on the mathematical intuition behind "...the resulting symmetric Euclidean distance matrix has a rank at most 5", but not simply have a reference there. If they want to have a broad audience, this is not quite a responsible way of doing things.

Response: Thanks for this great suggestion. We have now provided detailed explanations on the low-rank property and why the rank of the Euclidean distance matrix is at most 5, in the 1st paragraph of the section "Reconstruct 3D genome structures based on low-rank matrix completion" in Methods. In addition, a brief intuitive explanation is also provided in the 2nd paragraph of the section "FLAMINGO algorithm to reconstruct high-resolution 3D genome architectures" in Results.

2. Along the manuscript, the Hi-C map and distance map seemed to be used interchangeably. Although the authors provided evidence to show that the threshold of converting from distance map to contact map is insensitive, they need to be very clear on the concept that these two things are far away from each other [PMID: 25512564, PMID: 26223848, PMID: 27760553, PMID: 28604723]. One needs to be paying more attention especially when the resolution of the model is higher. I doubt the current model could capture any of these important observations, as the root between the inconsistency of the Hi-C contacts and spatial distance might come from the structural heterogeneity. But if the authors have anything else to show, it would be necessary to show to make this important distinction clear.

Response: As a clarification on our writing, in the manuscript, we use "Hi-C contact map" to refer to the matrix of Hi-C contact frequency, and use "distance matrix" to refer to the converted distance matrix from the Hi-C contact map. So they are not used interchangeably.

The conversion from contact frequency to distance matrix is one of the standard pre-processing steps in the literature, including the methods that we have compared in the manuscript. Based on our own evaluations, in addition to the robustness analysis on the conversion factor (Supplementary Fig. 4), we also showed the consistency between our predicted structures with the image-based single-cell structures (Figure 5c-d, Supplementary Fig. 17). Taken together, these results strongly support the validity and robustness of the conversion from Hi-C contact maps to distance matrices.

Also, it is actually pointed out in the paper suggested by the reviewer (PMID: 27760553) that: "... the majority of studies suggest that 3C-based techniques and 3D DNA FISH give concordant results over a wide range of genomic and spatial ranges...". The same paper also pointed out that the special case suggested by the reviewer (PMID: 25512564) may actually result from massive chromatin changes caused by the gene knockout.

In addition, the concordance between Hi-C and 3D FISH and the conversion from contact frequency to spatial distance have been widely justified by many studies,

including: *Rao et al. 2015 Cell*, *Nora et al. 2012 Nature*, *Wang et al. 2016 Science*, *Kalhor et al. 2012 Nature Biotechnology*, *Crane et al. 2015 Nature*, *Georgetti et al. 2014 Cell*, *Chaumeil et al. 2013 Nature Communications*, and *Brackley et al. 2016 Genome Biology*.

3. How do the authors treat the centromeric regions, where a large chunk of Hi-C data is missing, in terms of the modeling and algorithmic aspect?

Response: Thanks for this note. The centromere and telomere regions are excluded from the models, which is the same as have been done by all other methods.

We have now clarified this in the 4th paragraph in the section “FLAMINGO algorithm to reconstruct high-resolution 3D genome architectures” in Results and the 1st paragraph in the section “Reconstruct 3D genome structures based on low-rank matrix completion” in Methods.

4. In figure 1b (and figure S10), the authors showed a representative CTCF-mediate loop. However, this is far from enough. A more systematic characterization of how well/bad the model is able to recapitulate the CTCF-loop is needed.

Response: Thanks for the suggestion. We have added a systematic comparison of the predicted spatial distances between CTCF-associated loop anchors vs. the rigorous genomic-distance controlled random pairs. FLAMINGO predicts statistically significant shorter spatial proximity for CTCF-associated loop anchors, suggesting its good performance in recapitulating CTCF-associated loops. The new results are summarized into Supplementary Figure 1 (the new boxplot) and are discussed in the 4th paragraph in the section “FLAMINGO algorithm to reconstruct high-resolution 3D genome architectures” in Results.

5. On page 7 in the section “Benchmark performance based on simulated structures”, the authors were comparing the performance of the model with benchmarked simulated structures. This is a useful comparison, but I wonder would the performance be affected by the length of the chain? For example, regarding the longest chromosome - chr1 - which has 44k fragments in the current setting (why 44k but not 50k? a separate question below), would you still get such a good benchmark performance? It seems that the benchmark system in figure 2 is rather short, but it is important to benchmark on a higher level.

Response: Thanks for this suggestion. We have carried out more benchmark analysis based on simulated structures with different lengths (range from 300 to 1,000), and the accuracy is robustly high. The new results are summarized in the Supplementary Figure 6, and are discussed in the 1st paragraph in the section “Benchmark performance based on simulated structures”.

As a clarification, because FLAMINGO models the chromosomal structures at 2 scales (e.g. 1Mb domain-scale and 5kb intra-domain scale), and can efficiently assemble and align the structures together, the lengths of the simulated structures are already comparable to human chromosomes. For example, chr1 contains 230 1Mb segments and each domain segment contains 200 5kb bins.

6. On page 8 in the section “Superior reconstruction accuracy across diverse cell-types”, The authors claimed a high correlation (0.95) for the skeleton prediction. This number, though looks high, probably does not say anything. With the current resolution, it is easy to achieve this kind of number. If the authors could show a good agreement between the prediction and Hi-C in a sequence-dependent manner, especially at the very off-diagonal region, it is more convincing and informative than the numbers shown here.

Response: First of all, as a clarification, the high-correlation for the skeleton prediction is not the focus in our manuscript and that is why it is included into the Supplementary Figures. We actually showed extensive performance evaluations and comparisons at 5kb-resolution: Figure 3b, 3c, 3e-f, Supplementary Figures 12 & 13.

And thanks for the suggestion to look at off-diagonal distances. We have now carried out the performance evaluation based only on off-diagonal points at 5kb-resolution. FLAMINGO still demonstrates high accuracy similar to the all-points evaluations. Across the 6 cell-types, the Spearman correlations based only on off-diagonal points range from 0.42 to 0.50, except for HUVEC (correlation=0.32). (Note: the data quality of HUVEC is lower than other cell types. It has the lowest sequencing depth, which is <32% of the other five cell types. Compared to GM12878, HUVEC’s sequencing depth is only 17.8%.) We have added this result into the 2nd paragraph of the section “Superior reconstruction accuracy across diverse cell-types”.

In addition, we have also compared FLAMINGO’s performance with other methods, based only on off-diagonal points. FLAMINGO still shows the highest accuracy, which is summarized in Supplementary Figure 11.

7. At the same place, the authors claimed the numbers are “consistently higher than other methods”. I assume these “other methods” are a few ones later on mentioned (e.g. Hierarchical3DGenome and such), but I am seeing it correctly, they were not introduced at all till such point. It led to a lot of confusion when I was reading this.

Response: We have now introduced the names of other methods in Introduction, so that the readers know these methods from an early stage.

8. Again, in the same paragraph, the authors claimed that “FLAMINGO reconstructs clear loop structures for TADs and predicts short 3D distances for inter-TAD chromatin contacts”. However, from the Hi-C, it doesn’t seem that the Hi-C shows a visibly strong

signal of an off-diagonal contact indicating the loop, which was seen strongly in the predicted map (figure 3a map). Could the authors comment on this inconsistency?

Response: This is actually an advantage of FLAMINGO to identify clear inter-TAD contacts, where the original Hi-C only has vague signals. As justification to the clear loop structure from our prediction, the Capture-C experiment in the same cell-type (Jung et al. Nature Genetics, 2019) shows significant long-range chromatin interactions linking the two distal regions. We have expanded Supplementary Figure 10 to show the supporting evidence from Capture-C interactions.

9. On page 9, the authors compared the Spearman correlations between predicted and observed 3D distances and generated a number. Again, this number is less useful compared to a sequence-dependent correlation, as a good correlation score might mainly come from the diagonal values, but I guess most of the available models are good in near-diagonal prediction but the off-diagonal values are more important and more difficult to capture.

Response: As mentioned above, we have now carried out the performance evaluation based only on off-diagonal points at 5kb-resolution. FLAMINGO still demonstrates high accuracy similar to the all-points evaluations. Across the 6 cell-types, the Spearman correlations based only on off-diagonal points range from 0.42 to 0.50, except for HUVEC (correlation=0.32). (Note: the data quality of HUVEC is lower than other cell types. It has the lowest sequencing depth, which is <32% of the other five cell types. Compared to GM12878, HUVEC's sequencing depth is only 17.8%.) We have added this result into the 2nd paragraph of the section "Superior reconstruction accuracy across diverse cell-types".

In addition, we have also compared FLAMINGO's performance with other methods, based only on off-diagonal points. FLAMINGO still shows the highest accuracy, which is summarized in Supplementary Figure 11.

10. On page 10, the authors did cross-comparison between different methods in capturing loci proximity from experimental values and showed their method consistently predicts a small pair-wise distance than other ones. However, I didn't see any direct comparison between the authors' model and experimental values.

Response: As a clarification, the experimental values (ChIA-PET, Capture-C and SPRITE) are used as gold standards here to benchmark the performance and to compare different methods. So we used these orthogonal experiments to do the systematic performance comparison between FLAMINGO and other methods.

11. On the same page, the authors compared one enhancer-promoter pair. This itself is a non-trivial question. If the authors want to make a good claim on this, they need to do a more comprehensive characterization of E-P pairs (there are public datasets on this,

e.g. PMID: 27064255) globally instead of only showing one or two pairs and make such an important statement. For example, what about E-P pairs without CTCF anchors? Is there any difference between these with the ones with CTCF? And if their model is really powerful in capturing those, of all the E-P pairs, what is the false positive/false negative rate of prediction?

Response: As a clarification, before the specific examples, Figure 3e-g actually show the global quantitative analyses that FLAMINGO can efficiently predict short spatial distances for chromatin interactions from ChIA-PET, Capture-C and SPRITE, which are commonly used to annotate regulatory links. These global analyses, along with the examples, are sufficient to support our claim that FLAMINGO “provide high-resolution structural supports for long-range regulatory links between enhancers and promoters”.

We actually didn't claim to “predict” E-P pairs in the manuscript, which is a separate topic from the current paper.

12. On the same page in section “Advanced scalability for large-scale chromosome conformations”, given the authors' statement, I found it would interesting to show the model's capability at 1kb if the authors really want to demonstrate the power of the model as they claimed, given that 1kb is the highest resolution with the bulk Hi-C data available, and also as probably known by the authors, 5kb, though indeed very high, is in principle not too impressive as there are other studies already achieving structural predictions at similar levels. I acknowledge that 1kb might be the hallmark at this point given all experiments we have and can be computationally challenging based on what the authors claimed, but if it can be shown on a smaller genomic region, e.g. 40Mb (~40k fragments), which is still similar to what the authors are doing here in terms of the computational complexity, that would be more interesting and also kind of a requirement to fulfill the authors' claim on the superior of the model in dealing with data sparsity/noise and high efficiency.

Response: Thanks for this great suggestion. We have now made 1kb-resolution predictions for all chromosomes of the human genome in GM12878. For example, for the largest chr1, FLAMINGO can generate the complete 1kb-resolution chromosomal structure prediction within 25 hours using 200GB memory. The all-points Spearman correlations with measured Hi-C are ~0.4 and the intra-domain Spearman correlations are ~0.6, at 1kb-resolution. These results are now summarized in Supplementary Figure 2. The predicted 1kb-resolution structures are also available at FLAMINGO's GitHub repository.

13. Also, given that the current pipeline runs for only 42 mins for 5kb on chromosome 1 and the computational complexity is $O(N^2)$, if 1kb is used that would presumably be 25 times slower, which is still acceptable on things like HPCC if available. Anyway, I don't see any reason of not going into a higher resolution if the advantage of the model that the authors claimed are really significant to be marked.

Response: Please check the response above.

14. Similar to comment 10, I didn't see any direct comparison between model results and experiments (e.g. SPRITE) in multi-way interactions, besides cross-comparison of different computational methods. Also, in figure 4a, first panel, is it really significant (the authors marked *** on the top)? From what I see, the two bars are almost aligned to each other.

Response: As a clarification, SPRITE is used as the gold standard here for multi-way interactions, and we are not competing with the experimental technique of SPRITE. The predicted shorter spatial distances suggest FLAMINGO is consistent with SPRITE data.

We have double checked the statistical test for the first panel of Figure 4a, and confirm that it is significant ($p\text{-value} < 10^{-3}$, $n=302$, one-sided Wilcoxon test).

15. Similar to comment 2, on page 15, the authors wrote that “it suggests that the distance information derived from Hi-C contact frequency is consistent with the spatial configurations obtained from imaging techniques”. This wording itself is problematic. The authors need to be careful.

Response: Please check our response to comment 2. “Hi-C contact frequency” refers to the original Hi-C signals and “distance derived from ...” refers to the converted spatial distances from original Hi-C signals.

Based on the results shown in Figure 5c-d and Supplementary Figure 17, it suggests the spatial information inferred from Hi-C is consistent with the image-based spatial reconstructions. Considering the special cases where these two techniques may not agree with each other, we revised the sentence as “... is overall consistent with ...”.

16. In the section “Reference structure to interpret single-cell variabilities” on page 15, the authors compared the model results with clusters of single-cell structures and found a high similarity of the predicted consensus structure to the averaged cluster 2 structure. Do these results indicate that the consensus structure might only be the representation of one of the local minimums in the entire structural ensemble characterized by the energy landscape, as the chromatin folding has been shown to be very frustrated and glass-like [PMID: 31467267, PMID: 30089831]. This might be reflected from the perspective of optimization in the authors' method, or might be the heterogenous nature of the chromatin population that the consensus structure method is not able to capture. This is an important point and the authors need to be more deliberate.

Response: Thanks for this question. As a clarification, although the averaged cluster 2 structure shows higher similarity, all the other 4 clusters also show good similarity to the predicted structure by FLAMINGO, as shown in Supplementary Figure 17 and discussed in the manuscript. As shown in Supplementary Fig. 17, the prediction error is

largely smaller than the intrinsic standard deviation across single cells in each specific cluster.

And we agree that the heterogeneity of chromatin structures across cells is an important issue. This point is discussed in the 3rd paragraph of Discussion (about the suggestion of taking the complementary advantages of FLAMINGO and ensemble-structure methods) and the new 4th paragraph of Discussion (about the future developments of using single-cell Hi-C data).

17. In the section “Cross cell-type prediction of 3D structures”, a more quantitative way to reveal the subtle difference would be to look at the correlation between the prediction map- for example, is the correlation between the distance map of K562 (predicted) and K562 (experiment) is higher than the correlation between the distance map of GM12878 (experiment) and K562 (experiment), to show that the model is able to capture the difference between different experiments.

Response: Thanks for the suggestion. We originally planned to do the comparison as suggested but found out that it is actually not feasible in practice. To calculate correlations between the distance maps of GM12878 (experiment) and K562 (experiment), we need to use the intersection set of points that have measured data in both GM12878 and K562. But due to the high missing rates and the cell-type difference, the intersection set of points is very small (<1% points of the original contact map) and only contains pairs of regions with short range (median genomic distance <200kb). Therefore, the suggested comparison can not be implemented.

Our current results actually provide a series of quantitative evaluations to strongly support the performance of cross cell-type predictions, as shown in Figure 7b, c, d, e, and Supplementary Figure 20.

18. I wonder if it is ever possible to apply the model to inter-chromosomal contacts, as they are also available as Hi-C, but of course, worse in terms of the quality and appear to be noisier. I am not an expert in this subfield but there are multiple studies out there that enable the whole-genome structural reconstruction from different perspectives and different types of method [PMID: 26951677, PMID: 25961318, PMID: 33086041]. I feel the authors’ model could have potentials on this perspective to generate some impacts. And also, this would not only be a good extension of the model but also a reasonable validation against the authors’ statement on the superiority of the model.

Response: Thanks for this suggestion. One of the major barriers to infer inter-chromosomal structures is the lack of reliable conversion functions to transform the inter-chromosomal Hi-C contact frequency to spatial distances, although the conversion function for intra-chromosomal contacts have been reliably estimated and justified. We have now discussed this limitation issue in the new 4th paragraph of Discussion for future developments.

19. Lastly, I don't think the authors gave a good discussion. As pointed out by a handful of recent reviews on different methods in chromatin modeling [PMID: 31470090, PMID: 32320675, PMID: 34241389, PMID: 33253996], clear drawbacks exist for the consensus structure method in reconstructing the chromatin structures. For example, it is not possible to study the dynamics, and some basic experimental observations on dynamical looping and neighboring TAD interactions cannot be revealed. However, the authors did not provide any substantial discussion on these aspects. The impression I got after reading is that they claim the model is totally the "supreme" one than anything else, but as detailed above, that's probably not true.

Response: Thanks for this note. We agree that drawbacks exist for consensus structure methods, although they do provide useful reference structures as the starting points to further understand the dynamics and variabilities. Systematic modeling of the structural dynamics and variabilities requires efficient modeling of single-cell Hi-C data, which raises unprecedented challenges on high rates of missing data (e.g. the missing rate is already >99.99% even at 100kb-resolution). We have now added one new paragraph in Discussion (4th paragraph) to discuss the limitations and future developments, including the current limitation of directly applying FLAMINGO on single-cell Hi-C data.

Minor issues:

1. Some portions of the abstract need to be rewritten. For example, key algorithm and key model innovation should be highlighted. Though it was mentioned that "compressing inter-dependent Hi-C interactions to delineate the underlying low-rank structures in 3D space", the statement remains allusive in revealing the core algorithm of the implementation.

Response: Thanks for this comment. We have expanded this sentence by including the core technique, *i.e.* "low-rank matrix completion", into the abstract, which is the foundation to address all the challenges listed in the first sentence of the abstract.

2. The authors need to be more careful in their word choice. For example, please avoid using things like "whole human genome", as it is very easy to give people the impression that it is the real whole-genome involving inter-chromosomal interactions as well. As the authors might know, there is quite an extensive amount of studies over there as well. However, the model is far from a complete "human genome".

Response: Thanks for this note. We have now removed "whole human genome" in the manuscript, and replaced it by "all chromosomes in the human genome".

3. On page 3, the sentence "The accuracy of a predicted structure is mainly evaluated by its capability of recapitulating the pairwise distances between genomic loci observed

from Hi-C, such as correlations”. This is a bad sentence, I guess the “correlation” is simply the “evaluation”, but it is very confusing as the Hi-C contacts or cross-loci distance is also a “correlation”. The last three words are basically meaningless to be here. Also, this sentence (and other places in the manuscript) is scientifically incorrect: one cannot observe pairwise distance from Hi-C.

Response: We have revised this sentence to improve the clarity. We have also added one sentence in the same paragraph to define the notation of observed distances, which are the distances converted from the observed Hi-C contact frequency, to clarify this notation used in this paper.

4. Why chromosome 1 (~250Mb) with 5kb only has 44k fragments?

Response: As mentioned in the response to the major comment 3, the centromere and telomere regions are excluded from the models, which is a standard step as done also by other methods. We have now clarified this in the manuscript: 4th paragraph in the section “FLAMINGO algorithm to reconstruct high-resolution 3D genome architectures” in Results, and the 1st paragraph in the section “Reconstruct 3D genome structures based on low-rank matrix completion” in Methods.

Reviewer #4 (Expertise: ML/DL, genomics):

This paper introduces a computational algorithm, FLAMINGO, for 3D reconstruction of the genome from Hi-C contact maps. The novelty of the method over previous algorithms is that the objective function used to predict the 3D coordinates of the genome is low-rank constrained. Thus, solving a low-rank matrix completion formulation, FLAMINGO can perform faster reconstruction from a subset of Hi-C data. The paper first shows that FLAMINGO can accurately model the simulated data. Next, it compares the method to other baselines on real-world datasets across 6 different cell lines. It also demonstrates low runtimes for an increasing number of loci compared to other methods. The subsequent sections use a variety of orthogonal datasets to demonstrate that the reconstructions from FLAMINGO - (1) capture pair-wise and multi-way chromatin interactions (2) capture eQTLs, hQTLs, and TAD curvature (3) are consistent with single-cell structures (4) produce high-quality Hi-C maps. Finally, the paper presents integrative-FLAMINGO (iFLAMINGO), an extension of the work, to incorporate 1D epigenomic signals in the 3D reconstruction to prejudice accurate cross-cell-type predictions.

Overall, with its comprehensive set of results, the paper makes a compelling case for

the proposed method. FLAMINGO is faster and more accurate than the existing methods and can assist in a variety of downstream biological applications. However, some points/descriptions in the paper require clarifications (see below) for a better understanding of the work.

Response: We really appreciate the reviewer's comments on the novelty of our method. We are glad that the reviewer found our results to be comprehensive and compelling.

Comments:

The choice of the baseline methods and the evaluation metric is not very clear. While the paper includes some recent methods, others like SuperRec (Zhang et al.) were not included. The paper need not include all the methods for comparison but a description of the relevant state-of-the-art would be helpful. Also, some of the previous works have included metrics like RMSD in addition to Spearman correlation for evaluation. Is there a reason why those were not included in the current study? Is correlation sufficient to compare distance matrices that tend to have underlying structural properties?

Response: Thanks for this comment. We have now included an explanation about the criteria of method selection for the performance comparison analyses, in the 2nd paragraph in the section "Performance comparison based on experimental Hi-C data" in Methods. And a brief summary of the existing methods is listed in the 2nd paragraph of the Introduction. In general, other methods that are not included in the comparison analyses are mainly because they have been compared to other methods in the literature and perform less accurately than the ones that we have included, or because they can not practically converge at 5kb-resolution (due to bugs or speed issues).

We have now also expanded our method comparison analyses by further including SuperRec, ShNeigh and RPR (which have not been compared to other methods in the literature and can practically converge for at least some of the chromosomes at 5kb-resolution). So FLAMINGO is now compared with 6 methods in total, and still shows the best performance.

The expanded performance comparison results are now shown in the revised or new figures: **a)** Figure 3c and Supplementary Figure 11 & 12 (accuracy comparison benchmarked by Hi-C data); **b)** Figure 3e-g and Supplementary Figure 13 (accuracy comparison benchmarked by orthogonal chromatin interaction datasets of ChIA-PET, Capture-C and SPRITE); **c)** Supplementary Figure 14b (computational scalability comparison); **d)** Figure 6e and Supplementary Figure 18b-e (comparison on the capability of handling missing data). These new results are discussed in the corresponding paragraphs in the Results and Methods.

As a clarification about RMSD, to evaluate the accuracy of predicted structures, RMSD can be used only if we know the true 3D coordinates of the genomic bins. Therefore, RMSD has been mainly used to benchmark the performance using simulated structures, as has been done in some of the previous work (SuperRec and ShNeigh).

We have actually done the equivalent analysis, in the section “Benchmark performance using simulated structures”, by calculating the relative errors (Figure 2), which are proportional to the squared RMSD values (with specified structure length and normalized by L2-norm). But for performance evaluation using real-world Hi-C data, since the true 3D coordinates are not known and only the measured pairwise distances are available, calculating RMSD is not feasible and Spearman correlations are the commonly used metric to benchmark the accuracy, as has been used also by other methods (Hierarchical3DGenome, ShRec3D, GEM-FISH, SuperRec, ShNeigh and RPR).

For a reader not familiar with the 3D reconstruction literature, it is not clear how some of the components of the algorithm compare or connect with the previous literature. How is this work placed with respect to previous works? For example, the hierarchical setup of constructing domain-wise skeleton and filling in intra-domain structures - is it similar to (or different from) the one proposed in Hierarchical3DGenome paper (Trieu et al.)?

Response: Thanks for this important comment. The major unique component of FLAMINGO is to systematically incorporate the low-rank property of the distance matrix, which is the main reason for a series of performance advantages (improve accuracy, scalability, robustness to missing data and flexibility to integrate with 1D data), as discussed in the 3rd paragraph of Introduction and in the 2nd paragraph of the section “FLAMINGO algorithm to reconstruct high-resolution 3D genome architectures” in Results.

And our iterative assembly algorithm (Supplementary Figure 3) to rotate and align intra-domain structures along the skeleton is new and is different from other methods. In other methods that also provide the hierarchical setup (GEM-FISH and Hierarchical3DGenome), the alignments are either restricted to just the end-points of each domain or are substantially dominated by intra-domain contacts (*i.e.* the contributions from the long-range inter-domain contacts are much smaller). Our new assembly algorithm can iteratively align all domains and rotate along 3 directions, by utilizing the long-range inter-domain (off-diagonal) contacts (Supplementary Figure 3). We have added the explanation into the 3rd paragraph of the section “FLAMINGO algorithm to reconstruct high-resolution 3D genome architectures” in Results.

Since the reconstructed structures from FLAMINGO assign shorter 3D distances between anchors of chromatin interactions, it would be interesting to see if there are examples (like the one presented for KCNA2) where the baseline methods (like Hierarchical3DGenome) did not capture the close proximity of enhancer and promoter. This result would solidify the importance of FLAMINGO for the downstream investigation of promoter-enhancer interactions.

Response: Thanks for the suggestion. Actually, the examples that we showed (including the one for KCNA2 in Fig. 3h and the additional examples in Supplementary Fig. 13) do

have the property that FLAMINGO assigns shorter 3D distances than the baseline method (*i.e.* Hierarchical3DGenome).

We have now expanded the Supplementary Fig. 13 by also showing the predicted structures from Hierarchical3DGenome for the same examples as comparisons. Due to the space limitation for Figure 3, we incorporate the expanded version for the KCNA2 example also into Supplementary Figure 13.

As shown in the expanded Supplementary Figure 13 with comparisons, for all of the three examples, FLAMINGO better demonstrates the spatial proximity between enhancers and the target gene promoters, which may be useful for downstream analyses of enhancer-promoter links.

It is unclear why the Hierarchical3DGenome was not included as a baseline in the results presented for multi-way interactions, eQTLs, and reconstructions from single-cell (Figures 4 and 5)

Response: Thanks for this question. As a clarification, the analyses on multi-way interactions, eQTLs, structure curvatures, and comparison to single-cell structures are carried out to demonstrate the usefulness of the predicted 3D structures for a variety of interesting biological studies, beyond basic analysis using 2D contact maps.

Since we have systematically compared with other methods for all aspects of algorithmic performance (*i.e.* accuracy benchmarked by Hi-C, accuracy benchmarked by orthogonal datasets, computational scalability, and robustness to handle missing data), the superior performance of FLAMINGO is justified and supported. Inclusion of other methods into all the biological analysis sections will make the figures and supplementary figures much bigger and deviate from the biological focus of the corresponding sections.

Supplementary Figure 17 does not show the values of the objective function for different values of dt parameter, instead, it seems that the value of 0.05 was chosen based on the intra-domain correlation metric. Is this value for dt consistent across all the experiments and all the cell lines?

Response: As a clarification, the intra-domain correlation metric is one of the major performance benchmark metrics to evaluate the accuracy of the predicted structures compared to measured Hi-C data. And because the dt parameter mainly affects the detailed structural configurations between consecutive bins, the intra-domain correlation metric can directly reflect the effects of different values of dt on the accuracy of predictions when FLAMINGO is applied on real data. Therefore, the intra-domain correlation metrics are used to choose the optimal value of dt .

And we agree with the reviewer that it is important to show the consistency of this value across different experiments and cell lines. We have evaluated the optimal value of dt across different chromosomes and across the 6 different experimental datasets (from different cell lines). The results are now added to Supplementary Figure 21c,

which shows the value concentrates within a narrow range around 0.05. This new result further supports the selected value of dt .

I could not find any descriptions about the parameters that need to be tuned for the baseline 3D reconstruction methods and whether adequate tuning was performed for them before evaluating their performance with FLAMINGO.

Response: We applied the baseline methods based on their suggested parameters and settings, and there is no parameter that needs to be tuned based on different datasets. The detailed descriptions on the baseline method implementations can be found in Supplementary Notes and also in the 2nd paragraph of the section “Performance comparison based on experimental Hi-C data” in Methods.

Minor points:

Figure 3(c) compares performances for multiple chromosomes but Supplementary figure 9 presents results for only chromosome 21. Is this because of runtime constraints?

Response: Yes, this is correct. For some of the baseline methods, it took too much time to make predictions for multiple chromosomes across all the other five cell-types. So we only compared the performance on chr21 for all methods across the five cell-types.

\alpha notation in the Methods is used for two different entities - (1) free parameter for distance matrix calculation (2) index of DNA fragment pair. Please use a different notation for one of them.

Response: Thanks for this note. We have changed the notation of the conversion factor for distance matrix calculation as η . The notation in Supplementary Fig. 4 is changed accordingly.

The method section description is dense making it hard to follow. It would be useful to separate the equations from the text and provide the equation numbers.

Response: Thanks for this suggestion. We have now listed and numbered the major equations in separate lines to improve the clarify of the Methods section.

Reviewers' Comments:

Reviewer #1:

Remarks to the Author:

My comments have been fully addressed. I am very impressed with the additional effort the authors went through to study distant genetic associations.

Reviewer #2:

Remarks to the Author:

The authors have addressed my concerns regarding comparison with other methods and reproducible and reusable codebase.

Reviewer #3:

Remarks to the Author:

The authors have addressed most of my concerns and the revised manuscript has largely improved. I am particularly impressed at the effort they made to extend the model to 1kb resolution for chromosomes in GM12878, which indeed increases the soundness of the model. There are still a few issues that remain to be resolved before I can recommend its final publication in Nature Communication:

1. When the authors mentioned "a systematic comparison of the predicted spatial distances between CTCF-associated loop anchors vs. the rigorous genomic-distance controlled random pairs", could they provide the rigorous definition of "rigorous genomic-distance controlled random pairs"? To my knowledge, these controls cannot be randomly selected genomic pairs with the same genomic separation, but have to be defined carefully. For example, one way to do that is: if A-B is a CTCF pair and B-C is a control pair, which uses B (one CTCF) and C has the same genomic separation with B as that of B with A. While there are other possible ways of doing it, the authors need to provide rigorous definitions that make sense.
2. When talking about "cross cell-type prediction of 3D structures", the authors mentioned that "due to the high missing rates and the cell-type difference, the intersection set of points is very small" and "only contains pairs of regions with short range". "Therefore, the suggested comparison cannot be implemented." To my knowledge, the paper published in 2020 provides the distance map for the entire chromosome (though not as high resolution as 5kb). Is it possible to use the experimental data published over there to make this comparison, maybe with a simple processing into a coarsen resolution same as the experimental data? The point is that we need to make sure it is not an overclaim, and "cross cell-type prediction" is a strong claim that needs to be verified through the comparison of pred-exp vs. exp-exp.
3. I don't feel my concerns in Comment 19 were properly addressed. While I agree with the authors that "the consensus structures provide useful reference structures as the starting points to further understand the dynamics and variabilities", the authors should at least briefly comment on the limitations of the consensus structure in the discussion section, for example also in the 4th paragraph of the discussion, to acknowledge the trending perspectives on this methodology from the listed review papers. I suggest the authors properly cite the review papers (PMID: 32320675, PMID: 34241389, PMID: 31470090) I listed, and comment about this point together with the single-cell limitations in the discussion section.
4. Similarly, the addition of the 4th paragraph in the discussion is helpful but it still needs to be extended. For example, I think the authors should also at least briefly comment in this section on the shortcut on the exclusion of centromeric and telomeric regions in the current algorithm along with the comment of the lack of accurate estimation of inter-chromosomal contacts, as centromeric regions are shown to be important in regulating genome organization in previous computational studies like PMID: 26951677 and PMID: 33086041. In other words, the authors

discussed the limit on the inter-chromosomal extension in the 4th paragraph of the discussion section but I think the importance of centromeric regions should be cited and commented as “a component that should not be excluded if organizations for the whole genome are to be assembled” to show the acknowledgement of those previous works and enlighten one of the possible extensions of the algorithm.

In summary, while I find that the revised manuscript has improved, I must invite the authors to further polish their text, especially a more objective, thoughtful discussion on the model limitation, along the mentioned lines.

Reviewer #4:

Remarks to the Author:

Thank you for addressing all my comments. The additional results (Fig 3c and 3h) and descriptions have helped highlight the contribution of the work and support the claims. Given the expanded performance results, I agree that the inclusion of other methods for all the biological analysis sections is not necessary.

Responses to Reviewer Comments

We appreciate reviewers' thoughtful comments and suggestions to improve our manuscript. We have addressed all the comments as explained below in the point-to-point responses.

REVIEWER COMMENTS

Reviewer #1 (Remarks to the Author):

My comments have been fully addressed. I am very impressed with the additional effort the authors went through to study distant genetic associations.

Response: Thanks for the suggestions that have strengthened our manuscript, especially the analyses of distant genetic associations.

Reviewer #2 (Remarks to the Author):

The authors have addressed my concerns regarding comparison with other methods and reproducible and reusable codebase.

Response: We appreciate the reviewer's suggestions that have helped to improve our method comparison and codebase work.

Reviewer #3 (Remarks to the Author):

The authors have addressed most of my concerns and the revised manuscript has largely improved. I am particularly impressed at the effort they made to extend the model to 1kb resolution for chromosomes in GM12878, which indeed increases the soundness of the model. There are still a few issues that remain to be resolved before I can recommend its final publication in Nature Communication:

Response: We appreciate the reviewer's insightful comments that have helped to strengthen and clarify our manuscript, especially the 1kb-resolution reconstructions. We have addressed all the remaining questions as explained in details below.

1. When the authors mentioned "a systematic comparison of the predicted spatial distances between CTCF-associated loop anchors vs. the rigorous genomic-distance controlled random pairs", could they provide the rigorous definition of "rigorous genomic-distance controlled random pairs"? To my knowledge, these controls cannot be randomly selected genomic pairs with the same genomic separation, but have to be defined carefully. For example, one way to do that is: if A-B is a CTCF pair and B-C is a control pair, which uses B (one CTCF) and C has the same genomic separation with B as that of

B with A. While there are other possible ways of doing it, the authors need to provide rigorous definitions that make sense.

Response: We appreciate this germane comment. We have carried out an additional comparison based on the reviewer's suggestion, where the controls are restricted to the pairs with one CTCF anchor and one random anchor with the same genomic separation. The new results are added to Supplementary Figure 1 (boxplot, right). Based on this new comparison, the predicted spatial distances for CTCF loop anchors (CTCF-CTCF pairs) are still statistically significantly shorter ($p\text{-value}=5.21\times 10^{-4}$, one-sided Wilcoxon test).

We have also clarified the definition of the controls and the statistical tests in the 4th paragraph of the section "FLAMINGO algorithm to reconstruct high-resolution 3D genome architectures". As explained in the new writings, there are two cases of controls used for comparison: 1) pairs between a CTCF-anchor and a random anchor with the same genomic separation (CTCF-random pairs, as suggested by the reviewer), which shows a $p\text{-value}=5.21\times 10^{-4}$ (one-sided Wilcoxon test, Supplementary Figure 1 boxplot, right); and 2) pairs between random anchors with the same genomic separation (random-random pairs), which shows a $p\text{-value}=2.78\times 10^{-5}$ (one-sided Wilcoxon test, Supplementary Figure 1 boxplot, left);

2. When talking about "cross cell-type prediction of 3D structures", the authors mentioned that "due to the high missing rates and the cell-type difference, the intersection set of points is very small" and "only contains pairs of regions with short range". "Therefore, the suggested comparison cannot be implemented." To my knowledge, the paper published in 2020 provides the distance map for the entire chromosome (though not as high resolution as 5kb). Is it possible to use the experimental data published over there to make this comparison, maybe with a simple processing into a coarsen resolution same as the experimental data? The point is that we need to make sure it is not an overclaim, and "cross cell-type prediction" is a strong claim that needs to be verified through the comparison of pred-exp vs. exp-exp.

Response: Thanks for this pertinent suggestion. We have now carried out this analysis as suggested, based on a coarse resolution (50kb-resolution) so that there are enough measured data for the exp-exp comparison. The new results are now added as the new Supplementary Figure 19c.

As shown in Supplementary Fig. 19c, evaluated by the experimental distances derived from the K562 Hi-C map, the iFLAMINGO predicted distances (based on the predicted GM12878->K562 structure from iFLAMINGO) achieve a higher correlation with the basic experimentally derived spatial distances from K562 Hi-C (pred-exp correlation=0.62), while the exp-exp correlation based on the basic experimentally derived spatial distances from GM12878 Hi-C and K562 Hi-C is only 0.55, suggesting that cross cell-type predictions by iFLAMINGO can capture the cell-type specificity.

In addition, we have further decomposed the comparison into intra-domain comparison and off-diagonal comparison (Sup. Fig. 19c), where the pred-exp correlations are all higher than the exp-exp correlations.

We have now added new texts about these results in the 2nd paragraph of the section “Cross cell-type prediction of 3D structures”.

3. I don't feel my concerns in Comment 19 were properly addressed. While I agree with the authors that “the consensus structures provide useful reference structures as the starting points to further understand the dynamics and variabilities”, the authors should at least briefly comment on the limitations of the consensus structure in the discussion section, for example also in the 4th paragraph of the discussion, to acknowledge the trending perspectives on this methodology from the listed review papers. I suggest the authors properly cite the review papers (PMID: 32320675, PMID: 34241389, PMID: 31470090) I listed, and comment about this point together with the single-cell limitations in the discussion section.

Response: Thanks for this suggestion. We agree with the reviewer to further explain the limitation of consensus-structure approaches and the biological importance of modeling the dynamics and cell-to-cell variability of chromatin structures.

We have now added new writings in the 4th paragraph of Discussion, based on the reviewer's suggestion, to emphasize the limitation of consensus structures and the importance of dynamic structures, together with the single-cell Hi-C limitation. And all the listed review papers by the reviewer are cited in the new text. In addition, we have also cited a series of ensemble-structure prediction algorithms, to acknowledge the previous work that are helpful in exploring the dynamic structural variabilities.

4. Similarly, the addition of the 4th paragraph in the discussion is helpful but it still needs to be extended. For example, I think the authors should also at least briefly comment in this section on the shortcut on the exclusion of centromeric and telomeric regions in the current algorithm along with the comment of the lack of accurate estimation of inter-chromosomal contacts, as centromeric regions are shown to be important in regulating genome organization in previous computational studies like PMID: 26951677 and PMID: 33086041. In other words, the authors discussed the limit on the inter-chromosomal extension in the 4th paragraph of the discussion section but I think the importance of centromeric regions should be cited and commented as “a component that should not be excluded if organizations for the whole genome are to be assembled” to show the acknowledgement of those previous works and enlighten one of the possible extensions of the algorithm.

Response: Thanks for this great suggestion. We have now added new writings in the 4th paragraph of Discussion, as suggested by the reviewer, to highlight the limitation of excluding centromere and telomere regions. We have cited the previous studies as suggested to acknowledge the importance of centromere in regulating chromatin organizations, and we have commented on the need of future developments to overcome the limitation in order to reconstruct complete structures for the whole genome.

In summary, while I find that the revised manuscript has improved, I must invite the authors to further polish their text, especially a more objective, thoughtful discussion on the model limitation, along the mentioned lines.

Response: We appreciate the thoughtful comments, and we have followed all the pertinent suggestions.

Reviewer #4 (Remarks to the Author):

Thank you for addressing all my comments. The additional results (Fig 3c and 3h) and descriptions have helped highlight the contribution of the work and support the claims. Given the expanded performance results, I agree that the inclusion of other methods for all the biological analysis sections is not necessary.

Response: We appreciate the reviewer's comments that have helped to improve the clarity and the results of our manuscript.

Reviewers' Comments:

Reviewer #3:

Remarks to the Author:

The authors have addressed most of my concerns and the additional data added that were shown in the Figures S1, S17, S19, etc. as well as the newly added text are pertinent and useful.

Responses to Reviewer Comments

REVIEWER COMMENTS

Reviewer #3 (Remarks to the Author):

The authors have addressed most of my concerns and the additional data added that were shown in the Figures S1, S17, S19, etc. as well as the newly added text are pertinent and useful.

Response: Thanks for the suggestions that have helped to strengthen our manuscript. We appreciate that the reviewer found our revisions to be pertinent and useful.